# The impact of R&D factors flow and regional absorptive capacity on China's economic growth: Theory and evidence

Xiumin Li[1], Furong Liang[1]*, Yabin Pi[1], Diexin Chen[2]

1 School of Economics, Guangdong University of Technology, Guangzhou, Guangdong, China, 2 School of Economics, Jinan University, Guangzhou, Guangdong, China

* 2376260390@qq.com

**Data Availability Statement:** All relevant data are within the manuscript and its Supporting Information files.

## Abstract

Innovation is the source of economic growth. Innovation in a region comes from its own knowledge creation and knowledge spillovers from other regions. Previous studies showed that R&D factors flow benefits knowledge spillover, thereby promoting economic growth. But these studies ignored the impact of a region's knowledge-absorptive capacity on knowledge spillovers. Ignoring the impact of regional absorptive capacity means that the knowledge spillover from the same R&D factors flow is the same, clearly inconsistent with reality. This thesis analyzes the impact of R&D factors flow on economic growth and explores the moderating effect of regional absorptive capacity on the relationship between R&D factors flow and economic growth from theoretical and empirical perspectives. First, we construct a knowledge creation and diffusion model of the new economic geography, including regional absorptive capacity, and analyze the theoretical logic of the flow of R&D factors and regional absorptive capacity influencing economic growth. Second, we employ spatial econometric models to examine the impact of R&D factors flow and regional absorptive capacity on economic growth, utilizing panel data of 30 provinces in China from 2008 to 2021. The results demonstrated a spatial positive correlation between regional economic growth in China. The R&D factors flow could have significantly promoted not just a region's direct economic growth, but also the economic growth of surrounding regions via spatial spillover effects. Furthermore, the stronger the regional absorptive capacity, the greater the direct effects and spatial spillover effects of the R&D factors flow on economic growth. The novelty of this article is to introduce regional absorptive capacity into the theoretical model, refine the methodology for assessing regional absorptive capacity in empirical research, and examine its moderating effect between the inflow of R&D factors and regional economic growth. This article reveals that the positive impact of the inflow of R&D factors on spatial spillovers and economic growth varies depending on regional absorptive capacity. According to the conclusions above, enhancing regional absorptive capacity is equally important as facilitating the flow of R&D factors. Therefore, it is vital for a region to strengthen its absorptive capacity for new knowledge while promoting R&D factors flow. The study provides valuable policy insights for accelerating the flow of

**Funding:** This work was financially supported by the National Natural Science Foundation of China (grant numbers 72173032) (https://www.nsfc.gov.cn/english/site_1/index.html), and the Natural Science Foundation of Guangdong Province of China (grant numbers 2021A1515011958) (http://gdstc.gd.gov.cn/). The funders had no role in study design. But the financial support for research, including statistical advice, data collection, and translation advice, is based on the funder.

**Competing interests:** No potential conflict of interest.

innovation factors, enhancing regional absorptive capacity, and consequently promoting long-term sustainable economic development in the region.

## 1. Introduction

Economic growth has always been a hot topic in economics. China's economic growth has dramatically improved since the country's reform and opening-up policy. According to the National Bureau of Statistics, China's gross domestic product (GDP) reached 40 trillion yuan in 2010 and exceeded 120 trillion yuan in 2022. However, since 2008, China's economic growth rate has been declining, reaching nearly 3% in 2022. Therefore, finding a sustainable economic development path is particularly urgent and vital. As an inexhaustible driver of economic growth, innovation can facilitate sustainable development by enhancing total factor productivity, accelerating the optimization and upgrading of the economic structure, and reducing energy consumption and carbon emissions [1–3]. The hundreds of years of economic development in developed countries have shown that innovation is the key to promoting sustainable economic development. Neoclassical, endogenous, and new economic geography (NEG) theories emphasize the role of research and development (R&D) in determining economic growth. Therefore, innovation is crucial to achieve long-term sustainable economic development. Understanding the internal mechanism of innovation-driven economic growth holds considerable theoretical value and far-reaching significance.

To enhance innovation capabilities, China has implemented the innovation-driven development strategy at the core of the nation's overall development. R&D factors (primarily R&D personnel and R&D capital) are the key factors regarding participation in the innovation process, reflecting innovation capability and influencing innovation outcomes. Some scholars argued that R&D factors are essential to secure economic interests and develop innovative technologies [4–6]. Therefore, R&D factors are indispensable strategic resources to ensure the effective implementation of innovation strategies. For a given region, there are two primary sources of R&D factors. The first is the input and accumulation of R&D factors in this region, and the second is the flow of R&D factors into this region from other regions. The flow of R&D factors mainly includes two key components: interregional talent migration and the transfer of innovation funding. The inflow of R&D factors from other regions, in addition to directly promoting the receiving region's economic growth by increasing the supply of R&D factors in it, also indirectly promotes the economic growth of surrounding regions via the spatial knowledge spillover effect. However, the magnitude of the knowledge spillover effect of the inflow of R&D factors depends on the receiving region's capacity to absorb new knowledge. The stronger the absorptive ability, the greater the knowledge spillover effect, the stronger the innovation ability, and the faster the economic growth. Nieto and Quevedo [7] asserted that the entire knowledge spillover process cannot be completed without absorbing external knowledge [7]. Therefore, improving the receiving region's ability to absorb new knowledge is also a crucial means of promoting regional innovation and accelerating regional economic growth.

There has been valuable research on the relationship among the R&D factors flow, spatial knowledge spillovers, and economic growth. Fujita and Thisse [8] combined endogenous growth theory and NEG theory in a natural manner and proposed a knowledge creation and diffusion model, using it to analyze the impact of the migration of R&D workers and R&D sectors on regional economic growth. Their research shows that when an economy shifts from decentralization to agglomeration, innovation occurs at a much faster rate; so long as the

economic growth effect induced by agglomeration is large enough, those who remain in the periphery also benefit more than in a decentralized state [8]. Inspired by it, Bai, et al. [9] incorporated the R&D factors flow into the knowledge creation and diffusion model of Fujita and Thisse [8] and proved that the R&D factors flow can promote economic growth via spatial knowledge spillovers [9]. Furthermore, they empirically investigated the impact of R&D personnel flow and R&D capital flow on China's economic growth using spatial panel models in Mainland China during 2000–2013. They discovered that both R&D personnel flow and R&D capital flow play a significant role in promoting a region's economic growth and have considerable spatial spillover effects on other regions. Audretsch and Belitski [10] demonstrated that the R&D factors flow and knowledge spillover enable enterprises in the regions to acquire external knowledge, fostering innovation and economic growth [10]. Wang, et al. [11] used the data of China from 2008 to 2018 and empirically found that R&D personnel and R&D capital inflows have a significant positive impact on regional economic growth, but the inflow of R&D personnel into the region has a significant negative spatial spillover effect on surrounding areas [11]. Wan, et al. [12] argued that both R&D personnel flow and R&D capital flow have significantly promoted China's hi-tech industry innovation capabilities, while government intervention exerts a negative regulatory impact between R&D factor flow and hi-tech industry innovation capabilities [12]. However, existing research on the impact of R&D factors flow on spatial knowledge spillover and regional economic growth has overlooked the role of regional absorptive capacity. This implies that the knowledge spillover and economic growth resulting from the flow of the same amount of R&D factors are assumed to be uniform across all regions, which is not the case, thus providing an opportunity for this study to investigate.

The concept of knowledge absorptive capacity was first proposed by Cohen and Levinthal [13]. Subsequently, researchers extended this concept to regional levels, defining regional absorptive capacity as the integrated ability of a region to acquire, digest, transform, and apply external knowledge [14]. Existing research suggests that the stronger the absorptive capacity of a region, the greater its ability to acquire knowledge from neighboring regions, and the faster its economic growth. Borensztein, et al. [15] argued that the actual impact of knowledge spillovers on economic growth depends not only on the volume of spillovers but also on the absorptive capacity of the economic entities [15]. Jung and López-Bazo [16] demonstrated that the process of knowledge spillover promoting economic growth relies on regional absorptive capacity [16]. Kaneva and Untura [17] suggested that the lagging regions fail to experience growth from knowledge spillovers due to their limited absorptive capacity [17]. Ahmed, et al. [18] stated that absorptive capacity plays an important role in spreading knowledge and economic growth [18]. However, there are differences in the measurement indicators and impact mechanisms of regional absorptive capacity in existing research. In terms of measurement indicators, many studies have used human capital [15] or R&D expenditures [14, 19] to measure regional absorptive capacity. Yet, such indicators are overly simplistic and lack persuasiveness [14, 15, 19]. Few scholars have used economic level, degree of openness [20], or patents [21] to define regional absorptive capacity. However, these externality indicators are far from the definition of absorptive capacity. Nevertheless, Zahra and George [22] proposed that absorptive capacity is a multidimensional concept with four dimensions: knowledge acquisition, knowledge digestion, knowledge transformation, and knowledge application [22]. When analyzing the impact mechanism, two important extensions can be distinguished. The first one is based on Romer's economic growth model, which establishes a regional economic growth model including regional absorptive capacity and uses econometric methods to test the moderating role of regional absorptive capacity in the relationship between spatial knowledge spillovers and economic growth [15, 20]. The second one is based on the spatially augmented Solow model proposed by Ertur and Koch [23], which applies spatial econometrics to capture

the interactive effects of regional absorptive capacity and spatial knowledge spillovers on economic growth [16, 24, 25]. From this, it can be seen that research on the correlation between regional absorptive capacity, knowledge spillover, and economic growth is relatively abundant. However, these literatures have overlooked the fact that the flow of R&D factors is the cause of spatial knowledge spillover.

In summary, the existing literature has separately investigated the impact of R&D factors flow on spatial knowledge spillover and economic growth or the influence of regional absorptive capacity on spatial knowledge spillover and economic growth. However, there is a limited amount of research literature that comprehensively analyzes the interplay among these four variables within a unified framework. Consequently, this study combines the two types of research mentioned above into a unified framework. We theoretically introduce R&D factors flow and regional absorptive capacity into the knowledge creation and diffusion model of Fujita and Thisse [8], constructing a model that includes R&D factors flow, spatial knowledge spillover, regional absorptive capacity, and economic growth. Moreover, we empirically construct a comprehensive regional absorptive capacity index system based on Zahra and George's [22] four dimensions of opinion to measure the regional absorptive capacity of China's provincial regions. Then, we use China's 2008–2021 interprovincial spatial panel model to examine the relationship between R&D factors flow, spatial knowledge spillover, regional absorptive capacity, and economic growth.

Our study is distinguished from other studies and makes contributions to the literature in the following aspects. First, we introduce regional absorptive capacity into the augmented knowledge creation and diffusion model of Bai, et al. [9]. We analyze the relationship between regional absorptive capacity and R&D factors flow, knowledge spillovers, and economic growth. In existing research on the flow of R&D factors, scholars have regarded that the knowledge brought by the flow of R&D factors can automatically lead to spillovers. They have not considered the impact of a region's knowledge-absorptive capacity on knowledge spillover effects. This means that the spillover effects brought by the inflow of unit factors are the same for all regions, but in fact, they are different. Our findings indicate that varying regional absorptive capacities result in diverse knowledge spillovers and economic growth when the same quantity of R&D factors flow in, thereby aligning the knowledge creation and diffusion model more closely with reality. Second, we improve the methodology for measuring regional absorptive capacity, expanding Zahra and George's [22] four-dimensional measurement of absorptive capacity into a comprehensive index system consisting of 31 indicators. This enhancement ensures a more scientific and accurate assessment of regional absorptive capacity. Third, we consider the directionality of R&D factors flow, and from the perspective of R&D factors inflow, we employ spatial Durbin model (SDM) to analyze the spatial spillover effects of regional absorptive capacity and R&D factors inflow on economic growth, further taking into account the moderating role of regional absorptive capacity in this relationship. Fourth, different from Wan et al.'s [12] study, which focuses on the impact of R&D factors flow and government intervention on innovation capabilities and employs the spatial autoregressive (SAR) model to demonstrate this [12], our research uses the SDM model to examine the relationships between R&D factors flow, regional absorptive capacity, and China's economic growth. Methodologically, the SDM model allows for the decomposition of the direct effects and spatial spillover effects of the independent variable (R&D factors flow), leading to more accurate spatial estimates compared to the SAR model. The rest of this paper is structured as follows: Section 2 discusses the theoretical model. Section 3 is on regional absorptive capacity in China. Section 4 discusses empirical methods, and Section 5 explains empirical results. Section 6 concludes the paper and provides some implications.

## 2. Theoretical model

Following the ideas of Fujita and Thisse [8] and Bai, et al. [9], our thesis adds regional absorptive capacity into the model framework of the knowledge creation and diffusion model, constructing a theoretical model that includes the flow of R&D factors, regional absorptive capacity, and economic growth.

### 2.1. Basic model

Assume that there are two regions in a country, namely regions A and B, and three sectors, namely the traditional sector (T), manufacturing sector (M), and innovation sector (R). There are two production factors, namely unskilled workers (L) and the R&D factors (H). Both the T-sector and M-sector use unskilled workers, while the R-sector uses R&D factors. Each unskilled worker is immobile and endowed with one unit of L-labor per unit of time. Every region has the same number of unskilled labor over time; thus, L is constant. Each R&D factor can move between regions at positive cost. The total number of skilled workers in an economy is constant over time and this number is normalized to 1. The traditional sector uses unskilled workers to produce homogeneous products, the innovation sector uses R&D factors (j) to produce new knowledge (e.g., patents), and the manufacturing sector uses the knowledge produced by the innovation sector to manufacture differentiated products. Note that the R&D factors in this article refer to R&D personnel and R&D capital. Referring to the treatment method of Bai et al., the two R&D factors are not distinguished in the theoretical analysis because both R&D personnel and R&D capital with knowledge are knowledge sources.

The T-sector operates within the framework of Wallas, utilizing unskilled labor units to produce homogeneous products. The transaction of T-goods between regions is cost-less, thus, we normalize its price to 1. The wage rates for the sector T in regions A and B are equal, and we standardize them to 1. Now, we turn to the M-sector; in the M-sector, the M-sector follows the Dixit-Stiglitz analysis framework to produce differentiated products. Each M-firm produces a differentiated product, and the production function is $\Pi i + a_L$, where $\Pi_i$ is the unit price for this new knowledge and $a_L$ is the number of unskilled workers required. There is an "iceberg" transport cost in M-products trade between regions. New knowledge is a product manufactured by R&D factors and must be purchased from the R-sector, and consumers cannot directly purchase it. The R-sector is in a perfectly competitive market, where the prices of products are the same and all products are bought by the M-sector. The M-sector requires the R-product as a fixed production cost.

Let the total amount of knowledge in region $r \in \{A,B\}$ be $K_r$, the share of R&D factors in region $r$ is $\lambda_r$. Each specific R&D factor $j$ has a specific amount of personal knowledge given by $h(j)$. The total amount of knowledge in region $r$ consists of two parts: one is knowledge produced by local R&D factors and the other is knowledge spillover from other regions. Moreover, this part is determined by the amount of knowledge in other regions, degree of external spillover, and absorptive capacity of the receiving region. Then, the knowledge available in region $r$ is given by:

$$K_r = (\int_0^{\lambda_r} h(j)^\beta dj + \theta_r \eta_r \int_0^{1-\lambda_r} h(j)^\beta dj)^{\frac{1}{\beta}}, \tag{1}$$

where $\beta$ represents an inverse measure of R&D factors' complementarity in knowledge creation, reflecting the heterogeneity of R&D factors, $0 < \beta < 1$. The parameter $\eta_r$ ($0 \leq \eta_r \leq 1$) expresses the intensity of knowledge spillovers; the larger the $\eta_r$, the faster the growth of new knowledge creation in $r$ region. The parameter $\theta_r$ ($0 \leq \theta_r \leq 1$) expresses the absorptive capacity

of $r$ region to "understand" knowledge spillovers from other regions, the larger the $\theta_r$, the faster the growth of new knowledge creation in $r$ region.

As M-firms invest specific differential knowledge to produce heterogeneous M goods, the knowledge stock is positively correlated with the number of M-firms. Without losing generality, assume that $h(j) = M, \lambda_A \equiv \lambda, \lambda_B \equiv 1-\lambda$ [8]. Using Eq (1), the amount of new knowledge produced by R&D factors in regions A and B is given by:

$$k_A(\lambda) = [\lambda + \theta_A \eta_A (1 - \lambda)]^{\frac{1}{\beta}}, k_B(\lambda) = [1 - \lambda + \theta_B \eta_B (1 - \lambda)]^{\frac{1}{\beta}} \tag{2}$$

By taking partial derivatives of Eq (2) for the knowledge spillover and absorptive capacity, we get the following:

$$\frac{\partial k_A(\lambda)}{\partial \eta_A} = \frac{\theta_A(1 - \lambda)}{\beta}[k_A(\lambda)]^{1-\beta} > 0, \frac{\partial k_B(\lambda)}{\partial \eta_B} = \frac{\theta_B \lambda}{\beta}[k_B(\lambda)]^{1-\beta} > 0. \tag{3}$$

$$\frac{\partial k_A(\lambda)}{\partial \theta_A} = \frac{\eta_A(1 - \lambda)}{\beta}[k_A(\lambda)]^{1-\beta} > 0, \frac{\partial k_B(\lambda)}{\partial \theta_B} = \frac{\eta_B \lambda}{\beta}[k_B(\lambda)]^{1-\beta} > 0. \tag{4}$$

Eqs (3) and (4) are greater than 0, indicating that knowledge spillover and regional absorptive capacity positively promote regional knowledge's growth.

Then, we can obtain short-term equilibrium conditions and long-term equilibrium conditions (For specific details and settings, please refer to Fujita and Thisse [8] or Bai, et al. [9]). The short-term equilibrium conditions are:

$$M_A = \frac{E_A - \phi E_B}{(1 - \phi)E}M, M_B = \frac{E_B - \phi E_A}{(1 - \phi)E}M \tag{5}$$

$$\pi^* = \max\{\pi_A{}^*, \pi_B{}^*\} = \frac{\mu E}{\sigma M} \tag{6}$$

$$a_H = \prod \cdot M = \frac{\mu E}{\sigma} \tag{7}$$

In Eq (5), $M_A$ and $M_B$ represent the number of manufacturing firms in regions A and B, respectively. $E_A$ and $E_B$ represent the total expenditure of regions A and B, respectively. The total number of M-firms is $M \equiv M_A + M_B$. The total expenditure of the entire country is $E \equiv E_A + E_B$. $\pi_A$ and $\pi_B$ represent the profits of M-firms in regions A and B, respectively. $a_H$ is the asset value of all firms in the M-sector. $\Pi$ is the asset value of an individual M-firm (Based on Bai, et al. [9], we do not consider cross-period decision-making, so the asset value of an M-firm is equal to its short-term equilibrium profit). $\mu$ is the share of consumer consumption of goods in the M-sector. $\sigma$ is the constant substitution elasticity between different types of products in the M-sector. $\phi$ is the degree of free trade between regions.

The long-term equilibrium conditions are:

$$E_A = \frac{L}{2} + \lambda(1 + k_A)\frac{\mu E}{\sigma}, E_B = \frac{L}{2} + (1 - \lambda)(1 + k_B)\frac{\mu E}{\sigma}. \tag{8}$$

$$E = L/\{1 - \frac{\mu\lambda}{\sigma}[1 + k_A(\lambda)] - \frac{\mu(1 - \lambda)}{\sigma}[1 + k_B(\lambda)]\}, \tag{9}$$

where $E_A, E_B$, and $E$ are the total expenditure of regions A, B, and the entire country, respectively.

## 2.2. Spatial spillover effect of R&D factors flow, regional absorptive capacity, and economic growth

It may be assumed that R&D factors gather in region A, that is, $\lambda > 1/2$. As R&D factors are proportional to the number of M-firms, R&D factors flow satisfies the following Equation:

$$\Delta S_K = \frac{1}{2}[k_A(\lambda) - k_B(\lambda)] = \frac{1}{2}[\lambda + \theta_A \eta_A (1-\lambda)]^{\frac{1}{\beta}} - \frac{1}{2}[1 - \lambda + \theta_B \eta_B (1-\lambda)]^{\frac{1}{\beta}} \qquad (10)$$

In Eq (10), $\Delta S_K$ represents the number of R&D factors flowing from region B to region A, reflecting the excess of R&D factors in region A compared to region B. The partial derivative of spatial knowledge spillover on R&D factors flow can be obtained based on Eq (10) below:

$$\frac{\partial \eta_A}{\partial \Delta S_K} = \frac{2\beta}{(1-\lambda)\theta_A} k_A^{\beta-1} > 0, \frac{\partial \eta_B}{\partial \Delta S_K} = -\frac{2\beta}{\lambda \theta_B} k_B^{\beta-1} < 0. \qquad (11)$$

According to Eq (11), when $0 < \lambda < 1$ and when R&D factors flow from region B to region A, the spatial knowledge spillover from region B to region A continues to increase, but it harms region B.

By substituting Eq (2) into Eq (8), we can derive the impact of spatial knowledge spillovers on the total expenditure of the global economic system and the expenditure of region A as follows:

$$\frac{\partial E}{\partial \eta_A} = \frac{L \frac{\mu \lambda}{\sigma} \frac{\partial k_A(\lambda)}{\partial \eta_A}}{\{1 - \frac{\mu \lambda}{\sigma}[1 + k_A(\lambda)] - \frac{\mu(1-\lambda)}{\sigma}[1 + k_B(\lambda)]\}^2} \qquad (12)$$

$$\frac{\partial E_A}{\partial \eta_A} = \frac{\mu \lambda}{\sigma}[E \frac{\partial k_A}{\partial \eta_A} + (1 + k_A) \frac{\partial E}{\partial \eta_A}] \qquad (13)$$

When we combine Eq (12) with Eq (3), we get $\frac{\partial E}{\partial \eta_A} > 0$, and therefore $\frac{\partial E_A}{\partial \eta_A} > 0$. It indicates that the spatial knowledge spillover of region B to region A, $\eta_A$, has a positive impact on both the economic system's total expenditure and region A's expenditure.

Based on Eqs (11), (8), and (9), we can get the impact of R&D factors flow, $\Delta S_K$, on the expenditure of region A:

$$\frac{\partial E_A}{\partial \Delta S_K} = \frac{\partial \eta_A}{\partial \Delta S_K} \frac{\partial E_A}{\partial \eta_A} = \frac{2\mu \lambda L \{\sigma - \mu(1-\lambda)[1 + [1 - \lambda + \theta_B \eta_B (1-\lambda)]^{\frac{1}{\beta}}]\}}{\{\sigma - \mu - \mu \lambda[\lambda + \theta_A \eta_A (1-\lambda)]^{\frac{1}{\beta}} - (1-\lambda)[1 - \lambda + \theta_B \eta_B (1-\lambda)]^{\frac{1}{\beta}}\}^2} > 0. \qquad (14)$$

Eq (14) shows that the inflow of R&D factors from Region B to Region A positively affects Region A's economic growth. At the same time, from Eq (14), it can be seen that the absorptive capacity of region A, $\theta_A$, also has a positive impact on $\frac{\partial E_A}{\partial \Delta S_K}$; namely the larger the $\theta_A$, the smaller the $\{\sigma - \mu - \mu \lambda[\lambda + \eta_A \theta_A (1-\lambda)]^{1/\beta} - (1-\lambda)(1 - \lambda + \eta_B \theta_B \lambda)^{1/\beta}\}^2$, and the larger the $\frac{\partial E_A}{\partial \Delta S_K}$.

Above all, this study makes the following propositions:

**Proposition 1**: The inflow of R&D factors promotes the spatial knowledge spillover from the outflow region to the inflow region, increases the total amount of knowledge in the inflow region, and promotes the inflow region's economic growth.

**Proposition 2**: The absorptive capacity of the inflow region has a positive moderating effect on the economic growth effect of R&D factors inflow.

Next, we discuss an additional analysis of the above-mentioned research proposition. On the one hand, R&D factors are the carriers of knowledge, and their movement in an interregional space must be accompanied by the exchange, collision, and sharing of knowledge and

technology in different regions, resulting in innovation spatial knowledge spillover for knowledge and technology. The interregional diffusion of knowledge and technology is an unconscious, irrepressible spatial behavior with a high rate of return. The spillover of knowledge or technology generated by the R&D factors flow make it simpler for each region to share the benefits of knowledge and technology from external regional innovation systems, thereby reducing local innovation costs, enhancing regional production efficiency, and fostering local economic growth. On the other hand, the regional absorptive capacity influences the economic efficiency of the flow of R&D factors. Changes in regional absorptive capacity affect the economic efficiency of R&D factors flow if all other factors remain unchanged. In other words, absorptive capacity can modify the degree to which innovation factors impact economic growth. Different regions have different absorptive capacities, and the degree of utilization of external knowledge varies, influencing the economic growth effect brought by R&D factors flow. Under certain conditions, the stronger a region's absorptive capacity, the greater its ability to absorb and apply knowledge, and the greater the role of R&D factors flow in promoting economic growth.

## 3. Characteristics and facts of absorptive capacity of provincial-level regions in China

### 3.1. Calculation method of the absorptive capacity of China's provincial regions

This study refers to the four-dimensional absorptive capacity analysis framework of knowledge acquisition, knowledge digestion, knowledge transformation, and knowledge application proposed by Zahra and George [22]. It selects appropriate fundamental indicators and develops a comprehensive evaluation index system for regional absorptive capacity that includes 31 indicators. Table 1 presents the index system.

In the index system of regional absorptive capacity mentioned above, knowledge acquisition refers to a region's ability to acquire new external knowledge. For evaluation, we select relevant indicators reflecting the promotion of external new knowledge flow. Knowledge digestion refers to a region's ability to understand and comprehend new external knowledge. For evaluation, we select relevant indicators reflecting regional learning, understanding, and assimilation of new foreign knowledge. Knowledge transformation ability refers to a region's ability to produce new knowledge based on learning, understanding, and digesting new foreign knowledge. For evaluation, we select relevant indicators reflecting the results of new knowledge transformation. Knowledge application ability refers to the ability of a region to apply the new knowledge generated for the production of technology and product development. For evaluation, we select relevant indicators reflecting technology, product development, and its achievements and outputs.

### 3.2. Calculation and characteristic analysis of China's provincial regional absorptive capacity

Based on the above index system, we calculate the absorptive capacity of 30 provincial-level regions in China from 2008 to 2021. We use the data from the National Statistics Bureau website of China and *the* China Science *and* Technology Statistics Yearbook [26, 27]. The processing method for missing data is as follows. (1) The telephone penetration rates in 2008 and 2009 are obtained by summing the fixed telephone penetration rate and mobile phone penetration rate in the fourth quarter of that year, and the data are obtained from the statistics of the Ministry of Industry and Information Technology website of China [28]; (2) The number

**Table 1. Index system of regional absorptive capacity.**

| Evaluation dimension | Evaluation index | Unit | Evaluation dimension | Evaluation index | Unit |
|---|---|---|---|---|---|
| Knowledge acquisition | Line length of long-distance optical cable | km | Knowledge transformation | Total exports | dollars |
| | Mobile phone exchange capacity | units | | Total investment by foreign-invested enterprises | dollars |
| | Telephone (including mobile phone) penetration rate | % | | Administrative expenses of industrial enterprises above the designated size | yuan |
| | Internet broadband access ports | units | | Per capita disposable income of urban households | yuan |
| | The total length of postal routes | km | | Per capita consumption expenditure of residents | yuan |
| | Number of newspapers and periodicals issued | units | | Software business income | yuan |
| | The total amount of telecom business | yuan | | | |
| | Total postal business volume | yuan | | | |
| | Number of employed persons in the postal industry | person | | | |
| | Passenger turnover | km | | | |
| Knowledge digestion | Illiteracy rate of population aged 15 and above | % | Knowledge application | Technology market turnover | yuan |
| | The proportion of the population aged six and above with college education and above | % | | Full time R&D personnel in industrial enterprises above designated size | person |
| | Number of students in colleges and universities | person | | Internal R&D expenditures of industrial enterprises above designated size | yuan |
| | Number of faculty and staff in ordinary colleges and universities | person | | Number of R&D projects | unit |
| | Number of college graduates | person | | Number of new product items | unit |
| | Total number of libraries and museums | unit | | Funding for developing new products | yuan |
| | Total collection of public libraries | unit | | Sales revenue of new products | yuan |
| | | | | Number of domestic patent applications accepted | unit |

of R&D projects, number of new product items, funding for developing new products, and sales revenue of new products were missing in 2010. We replaced them with the average values of the corresponding indicators in 2009 and 2011.

According to the comprehensive indicator system in Table 1, we use the principal component analysis (PCA) method to calculate the absorptive capacity of various provinces in China (Tibet, Hong Kong, Macao, and Taiwan were ignored due to data limitations). The Kaiser-Meyer-Olkin index is 0.91, the Bartlett statistic is 10543.97, and the P value is significant at the 1% level; thus, it is suitable for PCA. Our PCA result received four principal components, with eigenvalues of 17.72, 4.74, 2.07, and 1.34, respectively. Their cumulative contribution rate is 0.83, and their uniqueness is less than 0.6. Therefore, they meet the selection criteria for principal components. Next, we use the "0–1 normalization method" [The formula is: $x_{ij} = (x_{ij} - min(x_{ij})) / (max(x_{ij}) - min(x_{ij}))$] to process the results of the PCA, making the value greater than 0 and reducing the influence of multicollinearity. Table 2 shows the final results.

Table 2 shows that from 2008 to 2021, the absorptive capacity of 30 provinces increased significantly. All provinces' absorptive capacities have increased by at least two times since 2008; Qinghai, Guizhou, Gansu, Hainan, Ningxia, Anhui, Yunnan, and Chongqing have all increased their absorptive capacity by more than five times. Simultaneously, the gap in absorptive capacity among provinces is very large and shows a decreasing trend. In 2008, 2015, and 2021, Guangdong Province had the highest absorptive capacity, with the values of 0.26, 0.53,

**Table 2. Calculation results of the absorptive capacity of 30 provinces in China from 2008 to 2021.**

| Area | 2008 | 2009 | 2010 | 2011 | 2012 | 2013 | 2014 | 2015 | 2016 | 2017 | 2018 | 2019 | 2020 | 2021 |
|---|---|---|---|---|---|---|---|---|---|---|---|---|---|---|
| Beijing | 0.24 | 0.26 | 0.29 | 0.33 | 0.37 | 0.41 | 0.43 | 0.45 | 0.48 | 0.50 | 0.55 | 0.60 | 0.63 | 0.68 |
| Tianjin | 0.11 | 0.12 | 0.13 | 0.15 | 0.17 | 0.18 | 0.19 | 0.20 | 0.22 | 0.24 | 0.26 | 0.27 | 0.28 | 0.31 |
| Hebei | 0.10 | 0.12 | 0.14 | 0.15 | 0.17 | 0.19 | 0.20 | 0.21 | 0.23 | 0.25 | 0.27 | 0.31 | 0.34 | 0.34 |
| Shanxi | 0.09 | 0.09 | 0.11 | 0.12 | 0.13 | 0.15 | 0.15 | 0.16 | 0.16 | 0.18 | 0.20 | 0.21 | 0.23 | 0.23 |
| Inner Mongolia | 0.06 | 0.08 | 0.11 | 0.12 | 0.14 | 0.15 | 0.16 | 0.17 | 0.18 | 0.19 | 0.21 | 0.22 | 0.22 | 0.23 |
| Liaoning | 0.14 | 0.16 | 0.18 | 0.19 | 0.22 | 0.24 | 0.24 | 0.26 | 0.25 | 0.26 | 0.27 | 0.29 | 0.30 | 0.31 |
| Jilin | 0.07 | 0.09 | 0.10 | 0.11 | 0.13 | 0.14 | 0.15 | 0.16 | 0.17 | 0.18 | 0.20 | 0.21 | 0.22 | 0.22 |
| Heilongjiang | 0.09 | 0.10 | 0.12 | 0.13 | 0.14 | 0.16 | 0.17 | 0.18 | 0.19 | 0.20 | 0.21 | 0.22 | 0.23 | 0.24 |
| Shanghai | 0.22 | 0.24 | 0.26 | 0.28 | 0.30 | 0.32 | 0.34 | 0.35 | 0.37 | 0.40 | 0.43 | 0.46 | 0.49 | 0.55 |
| Jiangsu | 0.21 | 0.25 | 0.30 | 0.33 | 0.39 | 0.43 | 0.45 | 0.47 | 0.50 | 0.55 | 0.60 | 0.67 | 0.73 | 0.75 |
| Zhejiang | 0.18 | 0.21 | 0.24 | 0.26 | 0.31 | 0.34 | 0.36 | 0.40 | 0.43 | 0.47 | 0.53 | 0.60 | 0.67 | 0.67 |
| Anhui | 0.04 | 0.07 | 0.10 | 0.12 | 0.14 | 0.16 | 0.17 | 0.19 | 0.20 | 0.23 | 0.26 | 0.28 | 0.32 | 0.34 |
| Fujian | 0.10 | 0.12 | 0.15 | 0.17 | 0.18 | 0.20 | 0.21 | 0.23 | 0.23 | 0.26 | 0.29 | 0.31 | 0.34 | 0.35 |
| Jiangxi | 0.07 | 0.08 | 0.09 | 0.10 | 0.12 | 0.13 | 0.14 | 0.15 | 0.16 | 0.18 | 0.21 | 0.24 | 0.26 | 0.27 |
| Shandong | 0.17 | 0.19 | 0.23 | 0.24 | 0.27 | 0.30 | 0.33 | 0.35 | 0.38 | 0.42 | 0.44 | 0.48 | 0.55 | 0.59 |
| Henan | 0.10 | 0.13 | 0.15 | 0.15 | 0.17 | 0.20 | 0.22 | 0.23 | 0.25 | 0.28 | 0.32 | 0.35 | 0.39 | 0.39 |
| Hubei | 0.11 | 0.12 | 0.15 | 0.16 | 0.18 | 0.21 | 0.23 | 0.24 | 0.26 | 0.28 | 0.31 | 0.33 | 0.35 | 0.37 |
| Hunan | 0.10 | 0.12 | 0.14 | 0.14 | 0.16 | 0.18 | 0.19 | 0.21 | 0.22 | 0.25 | 0.27 | 0.31 | 0.34 | 0.36 |
| Guangdong | 0.26 | 0.29 | 0.33 | 0.37 | 0.41 | 0.48 | 0.50 | 0.53 | 0.58 | 0.69 | 0.78 | 0.90 | 1.00 | 0.99 |
| Guangxi | 0.06 | 0.08 | 0.09 | 0.10 | 0.11 | 0.13 | 0.14 | 0.15 | 0.16 | 0.19 | 0.21 | 0.23 | 0.26 | 0.28 |
| Hainan | 0.02 | 0.03 | 0.05 | 0.06 | 0.07 | 0.08 | 0.08 | 0.09 | 0.10 | 0.11 | 0.13 | 0.14 | 0.18 | 0.21 |
| Chongqing | 0.05 | 0.07 | 0.09 | 0.10 | 0.12 | 0.13 | 0.15 | 0.16 | 0.18 | 0.20 | 0.22 | 0.24 | 0.26 | 0.28 |
| Sichuan | 0.10 | 0.13 | 0.15 | 0.17 | 0.19 | 0.23 | 0.25 | 0.27 | 0.28 | 0.31 | 0.35 | 0.39 | 0.43 | 0.42 |
| Guizhou | 0.01 | 0.02 | 0.05 | 0.06 | 0.06 | 0.08 | 0.09 | 0.09 | 0.10 | 0.13 | 0.15 | 0.17 | 0.19 | 0.20 |
| Yunnan | 0.04 | 0.04 | 0.08 | 0.08 | 0.09 | 0.11 | 0.12 | 0.13 | 0.14 | 0.16 | 0.18 | 0.21 | 0.23 | 0.22 |
| Shaanxi | 0.09 | 0.11 | 0.13 | 0.14 | 0.16 | 0.19 | 0.19 | 0.21 | 0.22 | 0.23 | 0.26 | 0.28 | 0.31 | 0.32 |
| Gansu | 0.01 | 0.03 | 0.06 | 0.07 | 0.08 | 0.10 | 0.10 | 0.11 | 0.12 | 0.14 | 0.15 | 0.16 | 0.18 | 0.18 |
| Qinghai | 0.01 | 0.01 | 0.04 | 0.04 | 0.05 | 0.06 | 0.07 | 0.06 | 0.07 | 0.09 | 0.11 | 0.11 | 0.12 | 0.12 |
| Ningxia | 0.02 | 0.03 | 0.04 | 0.05 | 0.06 | 0.07 | 0.07 | 0.08 | 0.09 | 0.10 | 0.11 | 0.11 | 0.13 | 0.14 |
| Xinjiang | 0.07 | 0.07 | 0.09 | 0.10 | 0.12 | 0.12 | 0.13 | 0.14 | 0.14 | 0.16 | 0.17 | 0.18 | 0.20 | 0.20 |

and 0.99, respectively, while Qinghai Province had the lowest absorptive capacity, with the values of 0.01, 0.06, and 0.12, respectively. Guangdong Province had 26 times, 8.83 times, and 8.25 times the absorptive capacity of Qinghai Province in 2008, 2015, 2021, respectively. The absorptive capacity of China's regions: Eastern, Central, and Western varies significantly. Eastern regions include Beijing, Tianjin, Hebei, Liaoning, Shanghai, Jiangsu, Zhejiang, Fujian, Shandong, Guangdong, and Hainan. Central regions encompass Shanxi, Jilin, Heilongjiang, Anhui, Jiangxi, Henan, Hubei, and Hunan. Western regions comprise Sichuan, Chongqing, Guizhou, Yunnan, Shaanxi, Gansu, Qinghai, Ningxia, Xinjiang, Guangxi, and Inner Mongolia. In 2008, 2015, and 2021, the absorptive capacity of the Eastern region measured 1.75, 3.55, and 5.75, respectively. Over the same years, the absorptive capacity of the Central region were 0.68, 1.53, and 2.43, respectively and those of the western region were 0.51, 1.55, and 2.59, respectively. Therefore, the difference between the absorptive capacity of the three major regions was considerably high. In summary, the significant disparity in regional absorptive capacity in China suggests that the effect of unit R&D factor inflows to regional economic growth varies depending on regional absorptive capacity.

## 4. Empirical methods

### 4.1. Spatial econometric model

**4.1.1. Spatial correlation test.** When testing the spatial autocorrelation relationship of a variable, calculating its Moran index is suitable. The formula for the Moran index is as follows:

$$Moran's\ I = \frac{n}{\sum_{i=1}^{n}\sum_{j=1}^{n}W_{ij}} \times \frac{\sum_{i=1}^{n}\sum_{j=1}^{n}W_{ij}(X_i - \bar{X})(X_j - \bar{X})}{\sum_{i=1}^{n}(X_i - \bar{X})^2}, \tag{15}$$

where $X_i$ is the economic development level of province $i$; $W_{ij}$ is the spatial weight matrix. The range of values for the Moran index is [–1,1]. At a given significance level, if the range of the Moran index is within (0,1], it means a positive correlation between regions.

### 4.1.2. Spatial weight matrix

Constructing an appropriate spatial weight matrix can help measure the spatial correlation between regions more accurately. This study chooses the spatial distance weight matrix; that is, the elements on the main diagonal are all 0, and the elements on the off-diagonal are $1/d$, where $d$ is the distance between the capital cities of two provinces. Its formula is as follows:

$$W^d = \begin{cases} 1/d, i \neq j \\ 0,\ other \end{cases} \tag{16}$$

### 4.1.3. Spatial econometric model

Anselin, Varga, and Acs [29] extend the production function using a spatial econometric model [29]. Spatial econometrics addresses estimation errors in regression when spatial data are involved [30]. Bai, et al. [9] shows that the flow of R&D factors among provinces is not independent of each other, and the flow of R&D factors to a certain province may be affected by the economic behavior of other provinces, and ignoring the spatial correlation accompanied by the flow of R&D factors may lead to the wrong setting of the model [9]. In addition, Ertur and Koch [23] show that there is also a spatial spillover effect on the knowledge absorptive capacity between regions [23]. Based on that, this study applies the spatial econometric model to examine the correlation between R&D factors flow, regional absorptive capacity, and economic growth. The model settings are as follows:

1. Spatial Durbin Model (SDM): The SDM model considers both a dependent variable's spatial correlation and the error term's spatial autocorrelation. The model is as follows:

$$\begin{aligned} \ln gdp_{it} &= \alpha_0 + \rho W \ln gdp_{it} + \alpha_1 \ln pf_{it} + \alpha_2 \ln cf_{it} + \alpha_3 \ln Z_{it} \\ &+ \theta_4 W \ln pf_{it} + \theta_5 W \ln cf_{it} + \theta_6 W \ln Z_{it} + \varepsilon_{it} \end{aligned} \tag{17}$$

To study the effect of regional absorptive capacity on the economic growth effect of R&D factors flow, this study adds the interaction term of absorptive capacity and R&D factors flow, based on Eq (17).

$$\begin{aligned} \ln gdp_{it} &= \beta_0 + \rho W \ln gdp_{it} + \beta_1 \ln pf_{it} + \beta_2 \ln cf_{it} + \beta_3 \ln ac_{it} + \beta_4 \ln ac_{it} \ln pf_{it} + \beta_5 \ln ac_{it} \ln cf_{it} + \beta_6 \ln Z_{it} \\ &+ \varphi_7 W \ln pf_{it} + \varphi_8 W \ln cf_{it} + \varphi_9 W \ln ac_{it} + \varphi_{10} W \ln ac_{it} \ln pf_{it} + \varphi_{11} W \ln ac_{it} \ln cf_{it} + \varphi_{12} W \ln Z_{it} + \varepsilon_{it} \end{aligned} \tag{18}$$

In Eqs (17) and (18), $t$ means time; $i$ means the region; $lngdp$ means economic development; $lnpf$ means R&D personnel flow; $lncf$ means R&D capital flow; $lnac$ means regional absorptive capacity; $Z$ means the control variables. Due to the selection of indicators related to human capital when calculating regional absorptive capacity, to avoid multicollinearity, the control variables in this study do not include human capital.

2. Spatial Autoregressive Model (SAR): The SAR model considers the spatial correlation of the dependent variable. The model is expressed as follows:

$$\ln gdp_{it} = \alpha_0 + \rho W \ln gdp_{it} + \alpha_1 \ln pf_{it} + \alpha_2 \ln cf_{it} + \alpha_3 \ln Z_{it} + \varepsilon_{it} \tag{19}$$

$$\ln gdp_{it} = \beta_0 + \rho W \ln gdp_{it} + \beta_1 \ln pf_{it} + \beta_2 \ln cf_{it} + \beta_3 \ln ac_{it} + \beta_4 \ln ac_{it} \ln pf_{it} + \beta_5 \ln ac_{it} \ln cf_{it} + \beta_6 \ln Z_{it} + \varepsilon_{it} \tag{20}$$

3. Spatial Error Model (SEM): The SEM model considers the spatial autocorrelation of the error term. The model is expressed as follows:

$$\ln gdp_{it} = \alpha_0 + \alpha_1 \ln pf_{it} + \alpha_2 \ln cf_{it} + \alpha_3 \ln Z_{it} + \mu_{it}, \mu_{it} = \lambda W \mu_{it} + \varepsilon_{it} \tag{21}$$

$$\ln gdp_{it} = \beta_0 + \beta_1 \ln pf_{it} + \beta_2 \ln cf_{it} + \beta_3 \ln ac_{it} + \beta_4 \ln ac_{it} \ln pf_{it} + \beta_5 \ln ac_{it} \ln cf_{it} + \beta_6 \ln Z_{it} + \mu_{it},$$
$$\mu_{it} = \lambda W \mu_{it} + \varepsilon_{it} \tag{22}$$

## 4.2. Variables and data sources

**4.2.1. Variables.** Gross Domestic Product ($lngdp$). We used the regional GDP converted into 2008 constant prices to measure the total regional output.

In our study, the R&D factors primarily refers to R&D personnel and R&D capital. Anderson [31] introduced the gravity model into the field of economics, and its general expression is as follows:

$$fl_{ij} = G_{ij} \cdot M_i \cdot M_j \cdot D_{ij}^{-2}, \tag{23}$$

where $fl_{ij}$ is the number of R&D factors flowing from region $j$ to region $i$; $G_{ij}$ is the gravity coefficient, usually taken as 1; $M_i$ and $M_j$ represent a specific scale of a region (e.g., population, capital); and $D_{ij}$ represents the distance between two areas. Based on the specific situation of the number of starting points and destination points, the gravity model can be divided into the following three forms: the full flow constraint gravity model that introduces the driving force variable and attraction variable; the output constraint gravity model that only introduces the attraction variable; the gravity model that only introduces the driving force variable.

The R&D personnel flow ($lnpf$). Labor economists often use the "push-pull" theory to explain the reasons for personnel turnover. According to this theory, population flow is the combined result of the pull factors in the inflow area and the push factors in the outflow area. Thus, when measuring the R&D personnel flow, this study considers wage as a primary driver affecting labor flow. Assuming the number of R&D personnel flowing from region $i$ to region $j$ is $pfl_{ij}$:

$$pfl_{ij} = \begin{cases} \ln P_i \cdot \ln(Wage_j - Wage_i) \cdot D_{ij}^{-2}, & Wage_j > Wage_i, \\ 0, & other \end{cases} \tag{24}$$

where $P_i$ is the number of R&D personnel in region $i$; $Wage_i$ and $Wage_j$ are the average wage of

urban unit employment in regions $i$ and $j$, respectively; $D_{ij}$ represents the distance from region $i$ to region $j$. Then, the total inflow of R&D personnel in region $i$ is:

$$pfl_i = \sum_j^n pfl_{ij} \tag{25}$$

The R&D capital flow (*lncf*). Due to the "profit-oriented" nature of R&D capital, its interregional flow mainly flows into regions with higher profits [32]. Thus, this study selects the profit rate in region $j$ as an attractive variable to measure R&D capital flow. Assuming the amount of R&D capital flowing from region $i$ to region j is $cfl_{ij}$:

$$cfl_{ij} = \ln C_i \cdot \ln Pfit_j \cdot D_{ij}^{-2}, \tag{26}$$

where $C_i$ is the R&D capital stock in region $i$; $Pfit_i$ and $Pfit_j$ represent the average profit rate in region $i$ and region $j$, respectively; $D_{ij}$ represents the same as above. Then, the total inflow of R&D capital in region $i$ is:

$$cfl_i = \sum_j^n cfl_{ij} \tag{27}$$

Regional absorptive capacity (*lnac*). The regional absorptive capacity is composed of 31 indicators. This thesis calculates its score using the PCA and "0–1 standardization" method. The comprehensive evaluation index system is presented in Table 1.

Labor (*lnL*). We measure the labor force by subtracting the number of R&D personnel from the total number of employees in society.

Capital (*lnK*). We measure capital by subtracting the value of R&D capital stock from the total physical capital stock. Both of them are calculated by the perpetual inventory method.

Transport Infrastructure (*lnbase*). We use the length of the railway operating mileage to measure the transportation infrastructure.

Market competition (*lnmkt*). Market competition is measured using the marketization comprehensive index published by the China Market Index Database.

Openness (*lnopen*). We measure openness by the proportion of foreign direct investment to the total regional output value.

## 4.2.2. Data sources

The R&D personnel and R&D capital data were sourced from the China Science and Technology Statistics Yearbook [27]. The telephone penetration rates data were sourced from the Ministry of Industry and Information Technology website of China [28]. The geographical distance data were sourced from China's National Platform for Common Geospatial Information Services [33]. The marketization comprehensive index data were sourced from the China Market Index Database [34]. The remainder of the data were sourced from the website of National Bureau of Statistics of China [26]. The handling method for missing data is as follows: (1) The telephone penetration rates for the years 2008 and 2009 are computed by summing the fixed-line telephone penetration rate and mobile phone penetration rate for the fourth quarter of each respective year. (2) For the year 2010, six indicators including full-time equivalent R&D personnel in industrial enterprises above designated size, internal R&D expenditures industrial enterprises above designated size, the number of R&D projects, the number of new product project item, funding for developing new products, and sales revenue of new products are represented as the average values of the corresponding indicators for the years 2009 and 2011. This research includes 30 provinces in China (excluding Tibet, Hong Kong, Macau, and Taiwan due to incomplete data) for the period 2008–2021. Table 3 presents the descriptive statistics.

Table 3. Descriptive statistical analysis of variables.

| Variable | Observations | Mean | Standard deviation | Minimum | Maximum |
|---|---|---|---|---|---|
| gdp | 420 | 20145.250 | 17492.270 | 896.900 | 99165.890 |
| pfl | 420 | 0.003 | 0.006 | 0.001 | 0.035 |
| cfl | 420 | 0.001 | 0.001 | 0.001 | 0.005 |
| ac | 420 | 0.222 | 0.153 | 0.001 | 1.000 |
| K | 420 | 58697.491 | 45507.513 | 1972.122 | 235839.100 |
| L | 420 | 2554.960 | 1637.011 | 277.005 | 7072.003 |
| base | 420 | 3792.772 | 2260.081 | 316.101 | 14209.499 |
| mkt | 420 | 7.825 | 1.889 | 3.359 | 12.399 |
| open | 420 | 11.454 | 59.712 | 0.917 | 1014.260 |

## 5. Empirical results

### 5.1. Spatial correlation results

We examined whether the provincial economic growth in China had spatial correlation. Table 4 reports the Moran index value and significance level of provincial economic growth in China during 2008−2021. According to Table 4, the Moran index values of economic growth during 2008−2021 were positive and statistically significant, indicating that provincial economic growth in China was not independent of neighboring regions but rather significantly influenced by them; that is, it had positive spatial correlation. Consequently, our study should use spatial econometric models for estimation.

### 5.2. Selection of the spatial panel model

Elhorst [35] considerd that the SDM is a more commonly used spatial model, but scholars should confirm that the SDM cannot degenerate into a SEM or SAR [35]. This study uses the

Table 4. Calculation results of Moran index.

| Year | Moran I | Standard deviation | T-value | P-value |
|---|---|---|---|---|
| 2008 | 0.0839*** | 0.0357 | 3.3159 | 0.0009 |
| 2009 | 0.0843*** | 0.0357 | 3.3289 | 0.0009 |
| 2010 | 0.0853*** | 0.0357 | 3.3573 | 0.0008 |
| 2011 | 0.0841*** | 0.0357 | 3.3259 | 0.0009 |
| 2012 | 0.0831*** | 0.0356 | 3.2984 | 0.0010 |
| 2013 | 0.0823*** | 0.0356 | 3.2787 | 0.0010 |
| 2014 | 0.0823*** | 0.0356 | 3.2793 | 0.0010 |
| 2015 | 0.0826*** | 0.0356 | 3.2858 | 0.0010 |
| 2016 | 0.0834*** | 0.0356 | 3.3074 | 0.0009 |
| 2017 | 0.0848*** | 0.0356 | 3.3469 | 0.0008 |
| 2018 | 0.0852*** | 0.0357 | 3.3536 | 0.0008 |
| 2019 | 0.0856*** | 0.0357 | 3.3654 | 0.0008 |
| 2020 | 0.0853*** | 0.0357 | 3.3582 | 0.0008 |
| 2021 | 0.0878*** | 0.0357 | 3.4271 | 0.0006 |

Note
*P < 0.1
**P < 0.05
***P < 0.01.

**Table 5. The LR and Wald tests' results.**

| Hypothesis | Model with no interaction term | | Model with an interaction term | |
|---|---|---|---|---|
| | LR-Value | Wald-Value | LR-Value | Wald-Value |
| H0: SDM can be simplified to SAR | 48.512*** | 56.016*** | 49.618*** | 56.669*** |
| H0: SDM can be simplified to SEM | 81.745*** | 90.862*** | 86.349*** | 96.791*** |

Note
*P < 0.1
**P < 0.05
***P < 0.01.

LR and Wald tests to examine whether the SDM can be simplified. In the LR test, the LR values had the 1% significance level and the H0 was rejected. In the Wald test, the Wald values also had the 1% significance level. These findings show that the SDM cannot degenerate to SEM or SAR. Therefore, the SDM is the most suitable model for our study. Table 5 lists the LR and Wald tests' results.

## 5.3. R&D factors flow, regional absorptive capacity, and economic growth in China

Table 6 reports the results of SDM, SAR, and SEM with space-time two-way fixed effects.

In Table 6, Columns (1), (3), and (5) exclude regional absorptive capacity, while Columns (2), (4), and (6) include regional absorptive capacity. The spatial rho or lambda values are significantly positive, indicating a positive spatial correlation between economic growth among provinces; that is, the surrounding regions' economic growth positively influences the economic growth of a region. The R-squared and log-likelihood statistics of the SDM model are larger than those of the SAR and SEM models, indicating that the SDM model cannot covert to the SAR or SEM models, regardless of whether the regional absorptive capacity is added. These results again show that this study should use the SDM model. Consequently, this article analyzes the results of the SDM model. However, Elhorst [35] noted that regression coefficients should be decomposed into direct and indirect effects before interpreting the regression results of an SDM model [35]. Table 7 shows the spatial effect decomposition result for Columns (1) and (2) of Table 6.

According to Model (1) in Table 7, the direct, indirect, and total effects of R&D personnel inflow and R&D capital inflow are all significant. Among them, the direct, indirect, and total effects of R&D personnel inflow reached the significance level of 1%, 10%, and 10% respectively, while the direct, indirect, and total effects of R&D capital inflow reached the significance level of 5%, 10%, and 10% respectively. The direct, indirect, and total effects of R&D personnel inflow are greater than the corresponding effect of R&D capital inflow. The indirect effect of R&D personnel and capital inflows is greater than their direct effect. The direct and indirect effects of R&D personnel inflow accounted for 12.121% and 87.878% of the total effect, respectively. Meanwhile, the direct and indirect effects of R&D capital inflow accounted for 9.302% and 89.583% of the total effect, respectively.

According to Model (2) in Table 7, after considering the regional absorptive capacity, the direct effect of the interaction term between R&D personnel inflow and regional absorptive capacity and the interaction term between R&D capital inflow and regional absorptive capacity are both positive and statistically significant. This demonstrates that regional absorptive capacity can enhance the impact of R&D factors flow on regional economic growth. The indirect effects of the interaction term between R&D personnel inflow and regional absorptive capacity

**Table 6. Spatial econometric estimation results.**

| | SDM | | SAR | | SEM | |
|---|---|---|---|---|---|---|
| | **(1)** | **(2)** | **(3)** | **(4)** | **(5)** | **(6)** |
| spatial rho or lambda | 0.534*** (4.854) | 0.502*** (4.312) | 0.765*** (13.196) | 0.747*** (12.251) | 0.708*** (9.433) | 0.675*** (8.090) |
| *lnpfl* | 0.020*** (3.974) | 0.022*** (4.631) | 0.018*** (4.240) | 0.013*** (3.400) | 0.017*** (4.043) | 0.012*** (2.992) |
| *lncfl* | 0.004** (2.002) | 0.005*** (3.108) | 0.004** (2.156) | 0.006*** (3.548) | 0.003* (1.899) | 0.005*** (3.282) |
| *lnac* | | 0.092*** (6.022) | | 0.103*** (7.445) | | 0.109*** (7.460) |
| *lnpfl lnac* | | 0.005*** (3.685) | | 0.004*** (3.330) | | 0.004*** (3.049) |
| *lncfl lnac* | | 0.008*** (5.285) | | 0.008*** (5.729) | | 0.007*** (4.867) |
| *W lnpfl* | 0.070* (1.695) | 0.124*** (3.164) | | | | |
| *W lncfl* | 0.017* (1.721) | 0.024** (2.390) | | | | |
| *W lnac* | | 0.181 (1.640) | | | | |
| *W lnpfl lnac* | | 0.031** (2.532) | | | | |
| *W lncfl lnac* | | 0.038*** (3.955) | | | | |
| Control variables | YES | YES | YES | YES | YES | YES |
| Individual fixed | YES | YES | YES | YES | YES | YES |
| Time fixed | YES | YES | YES | YES | YES | YES |
| R-squared | 0.987 | 0.987 | 0.986 | 0.986 | 0.949 | 0.953 |
| Log-likelihood | 869.763 | 887.581 | 845.489 | 855.706 | 828.910 | 836.728 |
| N | 420 | 420 | 420 | 420 | 420 | 420 |

Note: The numbers in parentheses represent T statistics.

*P < 0.10

**P < 0.05

***P < 0.01.

and the interaction term between R&D capital inflow and regional absorptive capacity are both positive and statistically significant, too. This shows that the regional absorptive capacity can enhance the effect of R&D factors flow on surrounding regions' economic growth.

## 5.4. Regional tests

Due to the apparent differences in R&D factors flow and regional absorptive capacity between regions, the impact of R&D factors flow and regional absorptive capacity on economic growth may be heterogeneous. Consequently, this study divides the country's 30 sample provinces into eastern, central, and western regions (the composition of these three regions has been explained in subsection 3.2.). Table 8 displays the SDM results of the regional tests.

According to Table 8, the impact of R&D factors inflow and regional absorptive capacity on economic growth varies considerably among eastern, central, and western regions. For the eastern region, Column (2) in Table 8 demonstrates that the direct effect of the interaction term between R&D capital inflow and regional absorptive capacity is not significant, the

**Table 7. Spatial effect decomposition results of the SDM model.**

| Model | Variable | Direct effect | | Indirect effect | | Total effect | |
|---|---|---|---|---|---|---|---|
| | | Coefficient | T-value | Coefficient | T-value | Coefficient | T-value |
| (1) | *lnpfl* | 0.024*** | 3.595 | 0.174* | 1.775 | 0.198* | 1.916 |
| | *lncfl* | 0.004** | 2.240 | 0.043* | 1.668 | 0.048* | 1.757 |
| (2) | *lnpfl* | 0.028*** | 4.097 | 0.280** | 2.378 | 0.308** | 2.495 |
| | *lncfl* | 0.006*** | 3.361 | 0.056** | 2.058 | 0.062** | 2.201 |
| | *lnac* | 0.103*** | 6.164 | 0.456* | 1.751 | 0.559** | 2.077 |
| | *lnpfl lnac* | 0.007*** | 3.864 | 0.070** | 2.242 | 0.076** | 2.363 |
| | *lncfl lnac* | 0.010*** | 5.614 | 0.088*** | 2.814 | 0.098*** | 3.040 |

Note

*$P < 0.10$

**$P < 0.05$

***$P < 0.01$.

indirect effect of the interaction term between R&D personnel inflow and regional absorptive capacity is not significant, and the effects of all other items are significantly positive. The direct effect of the interaction term between R&D personnel inflow and regional absorptive capacity in the eastern region is significantly positive, suggesting that the absorptive capacity of this region can amplify the effect of R&D personnel inflow on the economic growth of the inflow region. The significant positive indirect effect of the interaction term between R&D capital inflow and regional absorptive capacity in the eastern region indicates that the absorptive capacity of this region can enhance the effect of R&D capital inflow on the economic growth of neighboring regions.

For the central region, Column (4) in Table 8 demonstrates that that only the direct effects of R&D personnel inflow and R&D capital inflow, and the indirect effect of R&D personnel inflow, regional absorptive capacity, and the interaction term between R&D capital inflow and regional absorptive capacity are significant. This is very different from the overall sample regression results. Although the indirect effect of the interaction term between R&D capital inflow and regional absorptive capacity reaches a significance level of 10%, its effect is negative, which is contrary to the overall sample regression results. This result shows that for the central region, the more the R&D capital inflows from neighboring regions, the stronger the absorptive capacity, and the more unfavorable the region's economic growth. This may be because innovation in the central region is still in the agglomeration stage. Regions with more R&D factors are more attractive to R&D factors and have a greater capacity to absorb knowledge, creating a "siphon effect" that hinders the neighboring regions' economic growth. For the western region, Column (5) in Table 8 demonstrates that the direct and indirect effects of R&D personnel inflow are positive but not significant. This may be due to the relatively deficient resource endowment in the western region, which is insufficient to attract innovative talent. According to Column (6) in Table 8, the impact of all variables is significantly positive, which is generally the same as the overall sample.

## 5.5. Robustness test

This study employs two distinct methods to assess robustness. The first method involves replacing the spatial weight matrix. The estimations in Tables 6 and 7 are based on the negative power of distance to construct a spatial weight matrix, but different spatial weight matrices may cause different empirical results. We use the Queen adjacent matrix ($W^{01}$) and distance

**Table 8. Results of regional tests.**

| | Eastern region | | Central region | | Western region | |
|---|---|---|---|---|---|---|
| | (1) | (2) | (3) | (4) | (5) | (6) |
| Direct effect | | | | | | |
| *lnpfl* | 0.014** | 0.009 | 0.004*** | 0.006*** | 0.007 | 0.012*** |
| | (2.097) | (1.253) | (3.769) | (4.887) | (1.175) | (3.206) |
| *lncfl* | 0.055** | 0.057** | 0.005** | 0.006** | 0.008** | 0.017*** |
| | (2.001) | (2.136) | (2.183) | (2.331) | (2.204) | (3.297) |
| *lnac* | | 0.134*** | | 0.006 | | 0.020** |
| | | (3.537) | | (0.108) | | (2.094) |
| *lnpfl lnac* | | 0.017*** | | 0.001 | | 0.029*** |
| | | (3.914) | | (0.121) | | (4.050) |
| *lncfl lnac* | | 0.005 | | 0.016 | | 0.044*** |
| | | (0.457) | | (0.472) | | (3.639) |
| Indirect effect | | | | | | |
| *lnpfl* | 0.019 | 0.082* | 0.009*** | 0.015*** | 0.062 | 0.065*** |
| | (0.602) | (1.646) | (3.625) | (3.431) | (1.433) | (3.824) |
| *lncfl* | 0.333*** | 0.394*** | 0.005 | -0.002 | 0.060*** | 0.084*** |
| | (3.425) | (3.491) | (0.507) | (-0.167) | (2.955) | (6.540) |
| *lnac* | | 0.301** | | -0.788*** | | 0.081*** |
| | | (2.287) | | (-2.994) | | (3.059) |
| *lnpfl lnac* | | 0.031 | | 0.035 | | 0.144*** |
| | | (1.426) | | (1.097) | | (4.919) |
| *lncfl lnac* | | 0.118*** | | -0.222* | | 0.197*** |
| | | (3.243) | | (-1.910) | | (6.151) |
| Control variables | YES | YES | YES | YES | YES | YES |
| Individual fixed | YES | YES | YES | YES | YES | YES |
| Time fixed | YES | YES | YES | YES | YES | YES |
| Log-likelihood | 394.623 | 409.379 | 272.370 | 299.710 | 385.590 | 374.650 |
| R-squared | 0.937 | 0.987 | 0.993 | 0.976 | 0.347 | 0.608 |
| N | 154 | 154 | 112 | 112 | 154 | 154 |

Note: The numbers in parentheses represent T statistics.

*P < 0.10

**P < 0.05

***P < 0.01.

threshold matrix ($W^t$) [12, 36]. The setting method of Queen's adjacent matrix is: if two adjacent provinces have the same side or vertices, $w_{ij} = 1$; otherwise, $w_{ij} = 0$. The setting method of the distance threshold matrix is:

$$W^t = \begin{cases} w_{ij} = 0 & , i = j \\ w_{ij} = 0 & , i \neq j, d_{ij} > threshold. \\ w_{ij} = 1/d_{ij} & , i \neq j, d_{ij} < threshold. \end{cases} \quad (28)$$

where *threshold* represents the distance threshold, within the threshold is regarded as the adjacent provinces of province *i*, and the weight is a negative power between the distance between provinces *i* and *j*. The threshold of this article is set to 1906.380 kilometers (generated by ArcGIS). Columns (1) and (2) in Table 9 demonstrate the results of the Queen adjacent matrix, and Columns (3) and (4) in Table 9 show the results of the distance threshold matrix. The second method involves changing the measurement of the core variable. We adopt the approach

**Table 9. Robust test results.**

| | (1) | (2) | (3) | (4) | (5) | (6) | (7) | (8) | (9) |
|---|---|---|---|---|---|---|---|---|---|
| Direct effect | | | | | | | | | |
| lnpfl | 0.022*** (4.389) | 0.019*** (4.132) | 0.023*** (3.765) | 0.024*** (4.075) | 0.043*** (5.552) | 0.025*** (3.542) | 0.030*** (4.115) | 0.008** (2.295) | 0.009** (2.570) |
| lncfl | 0.004** (2.556) | 0.005*** (3.107) | 0.004** (2.064) | 0.005*** (2.851) | 0.010*** (3.828) | 0.005** (2.491) | 0.007*** (3.504) | 0.034** (2.464) | 0.026*** (4.811) |
| lnac | | 0.107*** (7.539) | | 0.093*** (6.088) | 0.090*** (6.143) | | 0.099*** (5.560) | | 0.015** (2.213) |
| lnpfllnac | | 0.005*** (3.983) | | 0.006*** (3.934) | 0.010*** (3.101) | | 0.008*** (3.903) | | 0.088*** (6.925) |
| lncfllnac | | 0.006*** (3.968) | | 0.006*** (3.980) | 0.034*** (3.330) | | 0.011*** (5.524) | | 0.006* (1.780) |
| Indirect effect | | | | | | | | | |
| lnpfl | 0.050*** (3.046) | 0.187** (2.560) | 0.148* (1.948) | 0.187** (2.560) | 0.439*** (3.030) | 0.181* (1.677) | 0.283** (2.204) | | |
| lncfl | 0.012** (2.234) | 0.026** (2.053) | 0.026* (1.853) | 0.026** (2.053) | 0.107*** (2.665) | 0.049* (1.655) | 0.057* (1.937) | | |
| lnac | | 0.218 (1.578) | | 0.218 (1.578) | 0.411* (1.940) | | 0.593** (1.978) | | |
| lnpfllnac | | 0.048** (2.246) | | 0.048** (2.246) | 0.126** (2.422) | | 0.085** (2.252) | | |
| lncfllnac | | 0.038*** (2.728) | | 0.038*** (2.728) | 0.356** (2.390) | | 0.093*** (2.626) | | |
| Control variable | YES | YES | YES | YES | YES | YES | YES | YES | YES |
| Individual fixed | YES | YES | YES | YES | YES | YES | YES | YES | YES |
| Time fixed | YES | YES | YES | YES | YES | YES | YES | YES | YES |
| Log-likelihood | 890.995 | 919.361 | 876.245 | 903.042 | 904.386 | 827.181 | 843.233 | 436.650 | 498.083 |
| R-squared | 0.977 | 0.984 | 0.979 | 0.991 | 0.988 | 0.985 | 0.987 | 0.959 | 0.979 |
| N | 420 | 420 | 420 | 420 | 420 | 420 | 420 | 420 | 420 |

Note: The numbers in parentheses represent T statistics.

*P < 0.10

**P < 0.05

***P < 0.01.

of Cohen and Levinthal [13] by using R&D intensity to measure absorptive capacity. The outcomes of this measurement can be observed in Column (5) in Table 9. Compared with Columns (1) and (2) in Table 6, although the coefficients are different, the direction has not changed fundamentally. These findings show that the estimation results mentioned above are robust.

There may be endogeneity issues with the estimation results in Table 6. First, the models likely overlook important variables influencing the economic growth in China. Second, R&D factors flow can affect China's provincial economic growth, while China's provincial economic growth can also affect R&D factors flow. This study conducts two additional tests to assess endogeneity. First, we perform SDM estimates using the lags of the core independent variables. Second, we use the Spatial Panel Autoregressive Generalized Method of Moments (SPGMM) for estimation to eliminate endogeneity issues [37]. Columns (6) and (7) in Table 9 show the results of the lag of core independent variables. Columns (8) and (9) in Table 9 demonstrate the results of the SPGMM. These findings again indicate the robustness of our study's results.

## 6. Conclusions and implications

This thesis analyzes the impact of R&D factors flow and regional absorptive capacity on economic growth from theoretical and empirical aspects. To analyze the theoretical logic of the R&D factors flow and regional absorptive capacity influencing economic growth, we expanded the knowledge creation and diffusion model of the NEG theory, which introduces regional absorptive capacity. Empirically, we calculated the knowledge absorptive capacity of 30 provinces in China from four dimensions. Using spatial panel models, we examined the impact of R&D factors flow and regional absorptive capacity on China's economic development and the moderating effect of regional absorptive capacity. The key findings are as follows. First, during the study period, the absorptive capacity of all regions of the country increased significantly. Compared with 2008, the absorptive capacity of all provinces increased by at least two times by 2021. However, the difference in absorptive capacity among provinces was also large. In 2008, 2015, and 2021, the absorptive capacity of the Eastern region measured 1.75, 3.55, and 5.75, respectively. Over the same years, the absorptive capacity of the Central region were 0.68, 1.53, and 2.43, respectively and those of the western region were 0.51, 1.55, and 2.59, respectively. The significant difference in regional absorptive capacity in China suggests that the effect of unit R&D factor inflows on regional economic growth will vary depending on regional absorptive capacity. Second, the Moran index and spatial item coefficient of the spatial econometric model demonstrated a significant positive spatial correlation and spatial knowledge spillover effect in provincial-level economic growth, indicating that the economic behavior of other regions influences economic activity in a region. Therefore, strengthening economic ties with neighboring regions contributes to a region's economic development. Third, the estimation results of the SDM demonstrated that R&D personnel flow and R&D capital flow not only significantly promote the economic growth of an inflow region but also significantly foster the economic growth of neighboring regions via the spatial spillover effect. The direct and indirect effects of R&D personnel inflow accounted for 12.121% and 87.878% of the total effect, respectively. Meanwhile, the direct and indirect effects of R&D capital inflow accounted for 9.302% and 89.583% of the total effect, respectively. Fourth, from the spatial perspective, the moderating effect of regional absorptive capacity is significant. Regional absorptive capacity has a positive moderating effect not only on the promoting effect of R&D factors flow on local economic growth but also on the promoting effect of R&D factors flow on the surrounding region's economic growth. According to the conclusions above, we find that enhancing regional absorptive capacity is equally important as facilitating the flow of R&D factors. The theoretical implication is that the knowledge brought by the flow of R&D factors would not automatically lead to spillovers. In fact, varying regional absorptive capacities result in diverse knowledge spillovers and economic growth when the same quantity of R&D factors flows in.

The policy implications are as follows. (1) The interregional R&D factors flow should be encouraged, thereby unleashing both the direct and spillover effects of such flows to bolster economic growth. In terms of stimulating the R&D personnel flow, the government should relax the restrictions on the settlement of talent, such as lowering the settlement threshold, increasing registered residence quotas, and expanding eligibility criteria to attract highly skilled individuals to migrate. Additionally, the government should reduce the settlement cost of talent, for instance, by increasing the supply of talent apartments and enhancing the relocation subsidies for incoming talent to address the housing problem of introduced skilled workers. In terms of fostering the R&D capital flow, the government should strengthen exchanges and cooperation in innovative activities among regions. For example, efforts should be made to incentivize cross-regional applications for open topics in national key laboratories and joint

applications by researchers from different regions for various levels and types of projects, including national and provincial natural science funds and social science funds. (2) Enhancing the ability of regions to absorb, transform, and utilize external knowledge, the moderating effect of regional absorptive capacity can be unleashed. This can be accomplished from three perspectives. Firstly, since education and human capital are important influencing factors of absorptive capacity, we can improve regional absorptive capacity by increasing investment in education, improving the quality of education, and enhancing human capital development, such as establishing high-level universities, strengthening the construction of teaching staff, and improving the education and teaching system. Secondly, the government should offer more favorable tax policies for scientific research and provide more R&D funding support for enterprises, and encourage their R&D investment. According to the dual role of enterprise R&D, this helps to improve the absorptive capacity of the enterprise, which in turn helps to strengthen the basic ability of the region to absorb foreign knowledge. Thirdly, efforts should be made to further improve transportation and information infrastructure, enhance interregional population, knowledge, and trade connections, creating more opportunities for knowledge inflow, thereby enhancing the region's ability to identify and absorb knowledge.

Next, we will engage in a discussion based on our conclusions. First of all, the overall findings of this study regarding the impact of the flow of R&D factors on economic growth through full-sample regression are generally consistent with Bai, et al. [9], suggesting that the flow of R&D factors has a promotional effect on the economic growth of the local region as well as neighboring regions. However, the analysis of regional heterogeneity in this study indicates that, in terms of the direct growth effects, the inflow of R&D factors into the eastern region leads to the greatest economic growth, followed by the western region, while the central region experiences the smallest gain. This pattern aligns closely with the respective absorptive capacities of these three regions. In other words, the impact of the flow of R&D factors is more significant in areas with higher absorptive capacity. In comparison, in areas with weaker absorptive capacity, the effect of the flow of R&D factors is less pronounced. In the second place, regarding the conclusions regarding spatial spillover effects, our study differs from Caragliu and Nijkamp's [14] research. Caragliu and Nijkamp's [14] study suggests that the greater the absorptive capacity of the local region, the smaller the knowledge spillover to the surrounding areas. In contrast, our research indicates that the greater the absorptive capacity of the local region, the larger the spatial spillover effects on the surrounding areas. This difference may be attributed to the choice of models in our study, which differs from the model chosen in Caragliu and Nijkamp's [14] research. Caragliu and Nijkamp's [14] study used specific indicators to measure outward knowledge spillovers and treated them as explained variables in the SAR model. In contrast, our spatial knowledge spillover was derived from the spatial effects decomposition of the SDM model, which was selected based on tests like LR and Wald.

Lastly, our study has limitations in measuring R&D factors flow. For the flow of R&D personnel, this study only selected wage levels as the primary influencing factor. However, it could also be influenced by other factors, such as environmental pollution levels [38]. Similarly, for the flow of R&D capital, this study only considered the average profit rate as the primary influencing factor. However, in reality, it could also be affected by the magnitude of investment risk [39]. While these limitations may affect the size of the estimated coefficients in the regression analysis, they do not change the conclusion that the inflow of R&D factors can positively impact the economic growth of the local region and its neighboring areas. Thus, future research could incorporate additional influencing factors that attract the flow of R&D factors and provide a more precise measurement of R&D factors flow.

## Supporting information

**S1 Dataset.**
(XLSX)

## Author Contributions

**Conceptualization:** Xiumin Li.

**Data curation:** Yabin Pi.

**Formal analysis:** Furong Liang.

**Writing – review & editing:** Diexin Chen.

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
