## [Decision Letter · Decision Letter 0]

2 Oct 2023

PONE-D-23-25669The impact of R&D factors flow and regional absorptive capacity on China’s economic growth: Theory and evidencePLOS ONE

Dear Dr. Liang,

Thank you for submitting your manuscript to PLOS ONE. After careful consideration, we feel that it has merit but does not fully meet PLOS ONE’s publication criteria as it currently stands. Therefore, we invite you to submit a revised version of the manuscript that addresses the points raised during the review process.

ACADEMIC EDITOR: Please revise as per reviewers comments. ==============================

We look forward to receiving your revised manuscript.

Kind regards,

Umer Shahzad, PhD

Academic Editor

PLOS ONE

“This work was supported by the National Natural Science Foundation of China [grant numbers 72173032](https://www.nsfc.gov.cn/english/site_1/index.html)

Yes, the fund is helpful.”

Reviewers' comments:

Reviewer's Responses to Questions

**Comments to the Author**

1. Is the manuscript technically sound, and do the data support the conclusions?

Reviewer #1: Yes

Reviewer #2: Yes

2. Has the statistical analysis been performed appropriately and rigorously? 

Reviewer #1: Yes

Reviewer #2: Yes

3. Have the authors made all data underlying the findings in their manuscript fully available?

Reviewer #1: Yes

Reviewer #2: Yes

4. Is the manuscript presented in an intelligible fashion and written in standard English?

Reviewer #1: Yes

Reviewer #2: No

5. Review Comments to the Author

Reviewer #1: It is my pleasure to review the manuscript for the esteemed journal. The manuscript studies the relationship between R&D factors flow and economic growth and explores the moderating effect of regional absorptive capacity on the relationship between R&D factors flow and economic growth from theoretical and empirical perspectives. The topic of the paper is important and worthy of investigation. Undeniably, the study could have contributed significantly if it had been developed, deepened, and broadened. However, the manuscript needs further improvement before it is accepted for publication. The reviewer has listed some specific comments that might be helpful for the author to enhance the quality of the manuscript further. Please consider the particular comments listed below.

Comment 1: Abstract. The abstract is well-written. However, it should further underscore the scientific value added to your paper in your abstract.

Comment 2: Sections of Introduction. The novelty of this paper should be further justified by highlighting the main contributions to the existing Introduction and Literature review. The application of emerging technologies can be seen as a typical R&D innovation. Please consider citing the following paper: entitled “Influence of artificial intelligence applications on total factor productivity of enterprises—evidence from textual analysis of annual reports of Chinese-listed companies.”; next entitled “Relationship between financial development and intelligent transformation of manufacturing: evidence from 69 countries.”, next entitled “The impact of AI on carbon emissions: evidence from 66 countries.”, and next titled “Impact of economic policy uncertainty shocks on China's financial conditions.”. There has already been a large number of literatures related to the research on the impact of innovation on the economy and society. Therefore, it should be better to elaborate on the contribution of the work to the existing literature.

Comment 3: The introduction section needs to give more prominence and clarity to the primary contributions, and it is recommended that the authors elaborate on this section and consider adding some additional contributions.

Comment 4: In the sections on Calculation and characteristic analysis of China’s provincial regional absorptive capacity, I would suggest that the authors show some of the elements of the principal component analysis measure in detail. For example, how many principal components were extracted? What is the cumulative contribution? This may be more convenient for readers to understand.

Comment 5: In the Robustness test section, it is unknown whether the authors can add ways such as changing the measurement of the dependent variable and the core explanatory variables to verify whether the same conclusion as the above empirical results can be obtained. Of course, this is a suggestion; I wonder if it's feasible for the author's reference.

Comment 6: The Conclusion and Policy Implications section should be pushed more towards discussing rather than summarizing the results. An in-depth discussion of the study's limitations and prospects for future research is welcome. Also, an indication of theoretical and practical implications would add significant value.

Good luck!!!

Reviewer #2: This paper investigates an interesting phenomenon about the impact of R&D factors flow and regional absorptive capacity on China’s economic Growth. The manuscript has been written well in large part. However, some revisions are required in this work.

1.The innovation of the article should be added in the abstract.

2.Abstract should include a brief statement about the policy implication.

3.I didn't see any innovation in the paper. The author should clearly indicate the innovation expenditure of the paper.

4.What is the connotation of R&D factors flow and regional absorptive capacity？

5.Since the analysis of the absorption capacity characteristics of provincial-level regions in China is involved, it is recommended to add spatial evolution analysis, which can be referred to in the following literature.

Yaping Xiao, Dalai Ma , Fengtai Zhang, et al. Spatiotemporal differentiation of carbon emission efficiency and influencing factors: From the perspective of 136 countries.[J].Science of the Total Environment.,2023,879:163032.

6.The data sources are too general, and different variables' data sources should be explained.

7.Why does the Moran Index value appear to be the same for multiple years? The reason should be explained.

8.Findings of the paper should be related to previous studies. albeit, some effort has been made in this regard, it is highly selective.

9.The policy recommendations are too general, and the feasibility and operability of the policy recommendations should be improved.

10. English expression needs further polishing.

11.The article should further point out the shortcomings and limitations.

6. PLOS authors have the option to publish the peer review history of their article (what does this mean?). If published, this will include your full peer review and any attached files.

Reviewer #1: No

Reviewer #2: No

---

## [Author Response · Author response to Decision Letter 0]

8 Nov 2023

Response to Reviewer 1 Comments

Reviewer #1: It is my pleasure to review the manuscript for the esteemed journal. The manuscript studies the relationship between R&D factors flow and economic growth and explores the moderating effect of regional absorptive capacity on the relationship between R&D factors flow and economic growth from theoretical and empirical perspectives. The topic of the paper is important and worthy of investigation. Undeniably, the study could have contributed significantly if it had been developed, deepened, and broadened. However, the manuscript needs further improvement before it is accepted for publication. The reviewer has listed some specific comments that might be helpful for the author to enhance the quality of the manuscript further. Please consider the particular comments listed below.

Letter to Reviewer #1

Dear Reviewer #1,

We sincerely thank you for your thorough review of our manuscript. We found them very useful to improve the quality of the manuscript. We fully appreciate your precious time and effort. Please accept our sincerest gratitude. We have responded to all your comments, and all revisions have been highlighted in the updated manuscript.

Warm regards,

Corresponding Author

Comment 1: Abstract. The abstract is well-written. However, it should further underscore the scientific value added to your paper in your abstract.

Response 1: Thank you for your recognition of this paper. Your insightful comments have been invaluable in enhancing this paper. Therefore, we have added the scientific value in the abstract. The details are as follows.

“

This article reveals that the positive impact of the inflow of R&D factors on spatial spillovers and economic growth varies depending on regional absorptive capacity.

”

See rows 47 to 48, or page 3 of the manuscript.

Thank you again for your valuable guidance and feedback.

Kind regards.

Comment 2: Sections of Introduction. The novelty of this paper should be further justified by highlighting the main contributions to the existing Introduction and Literature review. The application of emerging technologies can be seen as a typical R&D innovation. Please consider citing the following paper: entitled “Influence of artificial intelligence applications on total factor productivity of enterprises—evidence from textual analysis of annual reports of Chinese-listed companies.”; next entitled “Relationship between financial development and intelligent transformation of manufacturing: evidence from 69 countries.”, next entitled “The impact of AI on carbon emissions: evidence from 66 countries.”, and next titled “Impact of economic policy uncertainty shocks on China's financial conditions.”. There has already been a large number of literatures related to the research on the impact of innovation on the economy and society. Therefore, it should be better to elaborate on the contribution of the work to the existing literature.

Response 2: We sincerely thank you for your valuable comments. Our primary focus lies in examining the dynamic flow of R&D factors among regions from the perspective of factor flows, which differs from the perspective of the investment or accumulation of R&D factors. Therefore, after carefully reading the literature you provided, I have added them to the Introduction section of this manuscript. To clarify that innovation is very important and worth investigating. The details are as follows.

“

As an inexhaustible driver of economic growth, innovation can facilitate sustainable development by enhancing total factor productivity, accelerating the optimization and upgrading of the economic structure, and reducing energy consumption and carbon emissions [1-3].

”

See rows 62 to 65 of the manuscript.

I have added the fourth article to the section on prospects for future research in the article. (See the last paragraph of the manuscript.)

Next, existing literature on the relationship between R&D factors flow, spatial knowledge spillovers, and economic growth is limited, and we have identified only four relevant papers: Fujita and Thisse (2003), Bai et al. (2017), Audretsch and Belitski (2020), and Wang et al. (2021). Note that Wang et al. (2021) is a new addition to this revised manuscript.

“

Wang, et al. [8] used the data of China from 2008 to 2018 and empirically found that R&D personnel and R&D capital inflows have a significant positive impact on regional economic growth, but the inflow of R&D personnel into the region has a significant negative spatial spillover effect on surrounding areas [8].

”

See rows 107 to 111 or page 6 in the manuscript.

This addition serves to further highlight that the existing literature on the relationship between R&D factors flow, spatial knowledge spillovers, and economic growth has not considered a crucial variable: regional absorptive capacity.

The significance of studying R&D factors flow lies in the fact that the flow of R&D factors generates spatial knowledge spillovers, contributing additional value to the economy. However, regional absorptive capacity, as an influential variable affecting knowledge spillovers, undoubtedly impacts the process through which R&D factor flow affects economic growth via spatial knowledge spillovers. But existing literature has overlooked it.

Neglecting regional absorptive capacity implies that the same amount of R&D factor flow would result in the same knowledge spillovers and economic growth for all regions. However, in reality, different regions have varying capabilities to absorb external knowledge. Therefore, the inflow of R&D factors into different regions yields diverse outcomes in terms of knowledge spillovers and economic growth. Consequently, incorporating regional absorptive capacity into the model is reasonable and is worth exploring. We provide further elaboration on this matter in the Introduction section of the manuscript. The details are:

“

However, existing research on the impact of R&D factors flow on spatial knowledge 

spillover and regional economic growth has overlooked the role of regional absorptive 

capacity. This implies that the knowledge spillover and economic growth resulting from the flow of the same amount of R&D factors are assumed to be uniform across all regions, which is not the case, thus providing an opportunity for this study to investigate. 

”

See rows 111 to 115, or the last sentence of the third paragraph of the Introduction section, or page 6 in the manuscript.

Next, in order to demonstrate that existing literature on the relationship between knowledge spillovers, absorptive capacity, and economic growth has not explored the perspective of R&D factor flow, we have summarized the key points of Borensztein (1996) and Jung and López‐Bazo (2017) in the relevant section. Because these two articles are representative. Moreover, we added Kaneva and Untura (2019). This addition serves to further emphasize that the existing literature on the relationship between knowledge spillovers, absorptive capacity, and economic growth has not taken into consideration the aspect of R&D factor flow. The details are as follows.

“

Borensztein, et al. [11] argued that the actual impact of knowledge spillovers on economic growth depends not only on the volume of spillovers but also on the absorptive capacity of the economic entities[11]. Jung and López‐Bazo [12] demonstrated that the process of knowledge spillover promoting economic growth relies on regional absorptive capacity[12]. Kaneva and Untura [13] suggested that the lagging regions fail to experience growth from knowledge spillovers due to their limited absorptive capacity[13].

”

See rows 121 to 127, or page 7 in the manuscript.

Furthermore, we have added a corresponding statement in the concluding part of the fourth paragraph in the Introduction, emphasizing that existing literature on the relationship between knowledge spillovers, absorptive capacity, and economic growth has not considered the aspect of R&D factor flow. The details are as follows.

“

From this, it can be seen that research on the correlation between regional absorptive capacity, knowledge spillover, and economic growth is relatively abundant. However, these literatures have overlooked the fact that the flow of R&D factors is the cause of spatial knowledge spillover.

”

See rows 143 to 146, or the last sentence of the fourth paragraph of the Introduction section, or page 8 in the manuscript.

Please note that R&D factor flow and spatial knowledge spillovers are not the same; the former is one of the significant factors contributing to the latter.

Finally, in the fifth paragraph of the Introduction section, we have added two summarizing statements as follows.

“

In summary, the existing literature has separately investigated the impact of R&D factors flow on spatial knowledge spillover and economic growth or the influence of regional absorptive capacity on spatial knowledge spillover and economic growth. However, there is a limited amount of research literature that comprehensively analyzes the interplay among these four variables within a unified framework. 

”

See rows 147 to 151, or the first two sentences of the fifth paragraph of the Introduction section in the manuscript.

Consequently, this study combines the two types of research mentioned above into a unified framework.

What’s more, regarding the main contributions of this paper, you can refer to our response to your Comment 3.

Your expert guidance has indeed greatly enriched our paper, and we sincerely appreciate your advice.

Best regards!

Comment 3: The introduction section needs to give more prominence and clarity to the primary contributions, and it is recommended that the authors elaborate on this section and consider adding some additional contributions.

Response 3: We sincerely thank you for your valuable comments. Therefore, in the sixth paragraph of the introduction, we have reorganized and further elaborated on the innovative points of this article, and added an innovative point. The detailed content is as follows.

“

The main innovations of this article can be conveyed in three points. 

First, we introduce regional absorptive capacity into the augmented knowledge creation and diffusion model of Bai, et al. [6]. We analyze the relationship between regional absorptive capacity and R&D factors flow, knowledge spillovers, and economic growth. Our findings indicate that varying regional absorptive capacities result in diverse knowledge spillovers and economic growth when the same quantity of R&D factors flow in, thereby aligning the knowledge creation and diffusion model more closely with reality. 

Second, we improve the methodology for measuring regional absorptive capacity, expanding Zahra and George’s [17] four-dimensional measurement of absorptive capacity into a comprehensive index system consisting of 31 indicators. This enhancement ensures a more scientific and accurate assessment of regional absorptive capacity. 

Third, we consider the directionality of R&D factors flow, and from the perspective of R&D factors inflow, we employ spatial panel econometric models to analyze the spatial spillover effects of regional absorptive capacity and R&D factors inflow on economic growth, further taking into account the moderating role of regional absorptive capacity in this relationship.

”

See rows 161 to 175, or page 9 in the manuscript.

Kind regards.

Comment 4: In the sections on Calculation and characteristic analysis of China’s provincial regional absorptive capacity, I would suggest that the authors show some of the elements of the principal component analysis measure in detail. For example, how many principal components were extracted? What is the cumulative contribution? This may be more convenient for readers to understand.

Response 4: Undoubtedly, only reviewers well-versed in Principal Component Analysis could offer such professionally insightful suggestions. We greatly appreciate your input on this matter. Consequently, we have revised the relevant section of our paper about introducing Principal Component Analysis. The detailed content is as follows.

“

Our PCA result received four principal components, with eigenvalues of 17.72, 4.74, 2.07, and 1.34, respectively. Their cumulative contribution rate is 0.83, and their uniqueness is less than 0.6. Therefore, they meet the selection criteria for principal components.

”

See rows 357 to 360, or page 19, or the Calculation and characteristic analysis of China’s provincial regional absorptive capacity section of the manuscript for details.

Warm regards.

Comment 5: In the Robustness test section, it is unknown whether the authors can add ways such as changing the measurement of the dependent variable and the core explanatory variables to verify whether the same conclusion as the above empirical results can be obtained. Of course, this is a suggestion; I wonder if it's feasible for the author's reference.

Response 5: Without a doubt, only reviewers with a strong background in econometrics could provide such professional recommendations. Replacing core variables is one of the important robustness testing methods.

As regional absorptive capacity is our core variable, we followed Cohen and Levinthal’s (1990) approach and conducted a robustness check by using R&D intensity to represent absorptive capacity. The results confirm the robustness of our findings.

See rows Column (5) in Table 9, or pages 39 to 40 in the manuscript.

Note that, here, we no longer present the results without considering regional absorptive capacity because it is the same as Model (1) in Table 7.

We are very lucky to be able to obtain this result by effort.

Yours sincerely.

Comment 6: The Conclusion and Policy Implications section should be pushed more towards discussing rather than summarizing the results. An in-depth discussion of the study's limitations and prospects for future research is welcome. Also, an indication of theoretical and practical implications would add significant value.

Response 6: We sincerely appreciate your valuable feedback.

We have revised the previous general policy recommendations because they were not sufficiently detailed (See rows 691 to 712, or page 42 in the manuscript). 

In order to enhance the academic and practical value of our paper, we have conducted more discussions on the conclusions (See rows 713 to 734, or page 43 in the manuscript). 

We have provided a detailed introduction to the shortcomings of this article and potential future research directions (See the last paragraph of the manuscript).

Best wishes!!!

Response to Reviewer 2 Comments

Reviewer #2: This paper investigates an interesting phenomenon about the impact of R&D factors flow and regional absorptive capacity on China’s economic Growth. The manuscript has been written well in large part. However, some revisions are required in this work.

Letter to Reviewer #2

Dear Reviewer #2,

We sincerely thank you for your thorough review of our manuscript. We found them very useful to improve the quality of the manuscript. We fully appreciate your precious time and effort. Please accept our sincerest gratitude. We have responded to all your comments, and all revisions have been highlighted in the updated manuscript.

Warm regards,

Corresponding Author

1. The innovation of the article should be added in the abstract. 

Response 1: We sincerely thank you for your valuable suggestion. We agree to add the innovation points of our article in the abstract. The details are as follows.

“

The novelty of this article is to introduce regional absorptive capacity into the theoretical model, refine the methodology for assessing regional absorptive capacity in empirical research, and examine its moderating effect between the inflow of R&D factors and regional economic growth.

”

See rows 43 to 47 in the manuscript.

Your valuable suggestions have greatly helped to improve the quality of this article, and we are very grateful.

Kind regards.

2. Abstract should include a brief statement about the policy implication.

Response 2: We sincerely thank you for your valuable c

---

## [Decision Letter · Decision Letter 1]

5 Apr 2024

PONE-D-23-25669R1The impact of R&D factors flow and regional absorptive capacity on China’s economic growth: Theory and evidencePLOS ONE

Dear Dr. Liang,

Thank you for submitting your manuscript to PLOS ONE. After careful consideration, we feel that it has merit but does not fully meet PLOS ONE’s publication criteria as it currently stands. Therefore, we invite you to submit a revised version of the manuscript that addresses the points raised during the review process.

We look forward to receiving your revised manuscript.

Kind regards,

Umer Shahzad, PhD

Academic Editor

PLOS ONE

Journal Requirements:

**Additional Editor Comments:**

Dear authors,

The reviewers have provided the feedback on your manuscript. Based on these reports please revise the manuscript completely. In addition, please follow these comments; 

1. Please update the relevant literature review. 

2. Please add theoretical and policy level implications. 

3. Manuscript need editing, please ensure to edit the manuscript. 

4. The conclusion should be extended, and contribution should be justified.  

5. Contribution need to be extended. 

Reviewers' comments:

Reviewer's Responses to Questions

**Comments to the Author**

1. If the authors have adequately addressed your comments raised in a previous round of review and you feel that this manuscript is now acceptable for publication, you may indicate that here to bypass the “Comments to the Author” section, enter your conflict of interest statement in the “Confidential to Editor” section, and submit your "Accept" recommendation.

Reviewer #1: All comments have been addressed

Reviewer #3: (No Response)

2. Is the manuscript technically sound, and do the data support the conclusions?

Reviewer #1: Yes

Reviewer #3: Yes

3. Has the statistical analysis been performed appropriately and rigorously? 

Reviewer #1: Yes

Reviewer #3: Yes

4. Have the authors made all data underlying the findings in their manuscript fully available?

Reviewer #1: Yes

Reviewer #3: No

5. Is the manuscript presented in an intelligible fashion and written in standard English?

Reviewer #1: Yes

Reviewer #3: Yes

6. Review Comments to the Author

Reviewer #1: This revised version has been greatly improved. This paper shows the relationship between R&D and economic growth in emerging countries. This paper is of great scientific value. The current version already meets the requirements for journal publication.

Reviewer #3: This paper is within the scope of this journal and the methodology of the paper is also reliable. However, some areas can be improved before its publication. My suggestions for improvement are appended below:

Improve the abstract by adding an attractive background at the start of the paper. Also, include the main objective briefly, include period of investigation, and the sample of the study. Also, present the main findings followed by a sentence about policies.

Explain the advantages of the research methodology used in the context of this study. The appropriateness of the research methodology for this work should be provided.

The data section should be improved by following some international papers. All data sources must be referred in the text as well as in the reference list.

The contribution should be provided in a separate paragraph at the end of the introduction. Besides, this section should have a detailed background, the significance of the research, the justification of the sample, and the policy-related significance of the study.

In this section, conduct a critical evaluation of past literature, aiming to identify a literature gap that justifies the necessity of this study. This evaluation should be thorough, examining recent studies published within the last few years to ensure relevance and comprehensiveness. Below are some suggested studies to expand this section:

https://doi.org/10.1016/j.jclepro.2020.125529 0959-6526

https://doi.org/10.1007/s11356-021-17673-2

Improve the policy directions of the study.

7. PLOS authors have the option to publish the peer review history of their article (what does this mean?). If published, this will include your full peer review and any attached files.

Reviewer #1: No

Reviewer #3: No

---

## [Author Response · Author response to Decision Letter 1]

17 May 2024

Response to Reviewer 3 Comments

Letter to Review 3

Dear Review 3,

We sincerely thank you for your thorough review of our manuscript. We found them very useful to improve the quality of the manuscript. We fully appreciate your precious time and effort. Please accept our sincerest gratitude. We have responded to all your comments, and all revisions have been highlighted in the updated manuscript.

Warm regards,

Corresponding Author

Comment 1: Improve the abstract by adding an attractive background at the start of the paper. Also, include the main objective briefly, include period of investigation, and the sample of the study. Also, present the main findings followed by a sentence about policies.

Response 1: Thank you for your suggestions. We have added them in our abstract. The details are as follows:

(1) We have added the background at the start of the paper, which reads:

“

Innovation is the source of economic growth. Innovation in a region comes from its own knowledge creation and knowledge spillovers from other regions. Previous studies showed that R&D factors flow benefits knowledge spillover, thereby promoting economic growth. But these studies ignored the impact of a region’s knowledge-absorptive capacity on knowledge spillovers. Ignoring the impact of regional absorptive capacity means that the knowledge spillover from the same R&D factors flow is the same, clearly inconsistent with reality.

”

(2) We have added period of investigation, and the sample of the study in our abstract, which reads:

”

Second, … , utilizing panel data of 30 provinces in China from 2008 to 2021.

”

(3) We have presented the main findings followed by a sentence about policies, which reads:

“

According to the conclusions above, enhancing regional absorptive capacity is equally important as facilitating the flow of R&D factors. Therefore, it is vital for a region to strengthen its absorptive capacity for new knowledge while promoting R&D factors flow.

”

Kind regards.

Comment 2: Explain the advantages of the research methodology used in the context of this study. The appropriateness of the research methodology for this work should be provided.

Response 2: Thanks. We added them on page 24, section 4.1.3 of the manuscript, which reads:

“

Bai, et al. [9] shows that the flow of R&D factors among provinces is not independent of each other, and the flow of R&D factors to a certain province may be affected by the economic behavior of other provinces, and ignoring the spatial correlation accompanied by the flow of R&D factors may lead to the wrong setting of the model [9]. In addition, Ertur and Koch [22] show that there is also a spatial spillover effect on the knowledge absorptive capacity between regions[22]. Based on that, …

”

Kind regards.

Comment 3: The data section should be improved by following some international papers. All data sources must be referred in the text as well as in the reference list.

Response 3: Thanks. We checked the data section again and made some modifications, including listing them in the reference.

Please see page 19, section 3.2; page 28, section 4.2.2.; page 48, references 25,26,27,32,33.

Kind regards.

Comment 4: The contribution should be provided in a separate paragraph at the end of the introduction. Besides, this section should have a detailed background, the significance of the research, the justification of the sample, and the policy-related significance of the study.

In this section, conduct a critical evaluation of past literature, aiming to identify a literature gap that justifies the necessity of this study. This evaluation should be thorough, examining recent studies published within the last few years to ensure relevance and comprehensiveness. Below are some suggested studies to expand this section:

https://doi.org/10.1016/j.jclepro.2020.125529 0959-6526

https://doi.org/10.1007/s11356-021-17673-2

Response 4: Thanks. This suggestion is very helpful in improving the introduction section of our article. According to your suggestions, we improved the introduction section of our manuscript. 

First of all, we have provided our contributions at the end of the introduction. The details can be seen on page 9 of the manuscript. 

It is worth noting that we have expanded our contributions based on the comment#5 of the additional editor.

Secondly, we have expanded the introduction section part of our study. The specific modifications we have made are as follows:

We have added a reference on page 5, which reads: 

“Some scholars argued that R&D factors are essential to secure economic interests and develop innovative technologies[4-6].

”

Below are the literatures that we added:

doi: 10.1016/j.techfore.2019.04.007.

doi: 10.1007/s11356-021-17673-2.

doi: 10.1371/journal.pone.0299697.

These literatures are relevant to emphasize the necessity of the R&D factors.

We have added a reference on page 7, which reads: 

“

Ahmed, et al. [17] stated that absorptive capacity plays an important role in spreading knowledge and economic growth[17].

”

Below is the literature that we added:

doi: 10.1016/j.jclepro.2020.125529.

The addition of this reference helps to emphasize that studying the effect of regional knowledge absorption capability on knowledge spillover is of great importance.

Kind regards.

Comment 5: Improve the policy directions of the study.

Response 5: Thanks. We have improved the policy directions of our study. The details are as follow:

“

The policy implications are as follows. (1) The interregional R&D factors flow should be encouraged, thereby unleashing both the direct and spillover effects of such flows to bolster economic growth. In terms of stimulating the R&D personnel flow, the government should relax the restrictions on the settlement of talent, such as lowering the settlement threshold, increasing registered residence quotas, and expanding eligibility criteria to attract highly skilled individuals to migrate. Additionally, the government should reduce the settlement cost of talent, for instance, by increasing the supply of talent apartments and enhancing the relocation subsidies for incoming talent to address the housing problem of introduced skilled workers. In terms of fostering the R&D capital flow, the government should strengthen exchanges and cooperation in innovative activities among regions. For example, efforts should be made to incentivize cross-regional applications for open topics in national key laboratories and joint applications by researchers from different regions for various levels and types of projects, including national and provincial natural science funds and social science funds. (2) Enhancing the ability of regions to absorb, transform, and utilize external knowledge, the moderating effect of regional absorptive capacity can be unleashed. This can be accomplished from three perspectives. Firstly, since education and human capital are important influencing factors of absorptive capacity, we can improve regional absorptive capacity by increasing investment in education, improving the quality of education, and enhancing human capital development, such as establishing high-level universities, strengthening the construction of teaching staff, and improving the education and teaching system. Secondly, the government should offer more favorable tax policies for scientific research and provide more R&D funding support for enterprises, and encourage their R&D investment. According to the dual role of enterprise R&D, this helps to improve the absorptive capacity of the enterprise, which in turn helps to strengthen the basic ability of the region to absorb foreign knowledge. Thirdly, efforts should be made to further improve transportation and information infrastructure, enhance interregional population, knowledge, and trade connections, creating more opportunities for knowledge inflow, thereby enhancing the region’s ability to identify and absorb knowledge.

”

Please see pages 42 to 44 for details.

We sincerely thank you again for your valuable comments, which have been very helpful in improving our article.

Besh wishes!

Wishing you happiness every day!

 

Response to Additional Editor

Letter to Additional Editor

Dear Additional Editor,

We sincerely thank you for your thorough review of our manuscript. We found them very useful to improve the quality of the manuscript. We fully appreciate your precious time and effort. Please accept our sincerest gratitude. We have responded to all your comments, and all revisions have been highlighted in the updated manuscript.

Warm regards,

Corresponding Author

Comment 1. Please update the relevant literature review.

Response 1: We have diligently conducted another thorough review of the recent literature about our research topic, ensuring both the relevance and comprehensiveness of the references cited in our manuscript. The specific modifications we have made are as follows:

We have added a reference on page 5, which reads: 

“Some scholars argued that R&D factors are essential to secure economic interests and develop innovative technologies[4-6].”.

These literature are relevant to emphasize the necessity of R&D factors.

We have added a reference on page 7, which reads: 

“Ahmed, et al. [17] stated that absorptive capacity plays an important role in spreading knowledge and economic growth[17].”.

The addition of this reference helps to emphasize that it is important to study the regional knowledge absorption capability.

Kind regards.

Comment 2. Please add theoretical and policy level implications.

Response 2: We sincerely thank you for this valuable comment. We have added the theoretical implications of this study. Moreover, we have improved the policy implications of this study.

First of all, for theoretical level implications, we have added that:

“

The theoretical implication is that the knowledge brought by the flow of R&D factors would not automatically lead to spillovers. In fact, varying regional absorptive capacities result in diverse knowledge spillovers and economic growth when the same quantity of R&D factors flows in.

”

Please see it on page 42, the last sentence of section 6 of the manuscript.

Secondly, we have improved our policy implications:

“

The policy implications are as follows. (1) The interregional R&D factors flow should be encouraged, thereby unleashing both the direct and spillover effects of such flows to bolster economic growth. In terms of stimulating the R&D personnel flow, the government should relax the restrictions on the settlement of talent, such as lowering the settlement threshold, increasing registered residence quotas, and expanding eligibility criteria to attract highly skilled individuals to migrate. Additionally, the government should reduce the settlement cost of talent, for instance, by increasing the supply of talent apartments and enhancing the relocation subsidies for incoming talent to address the housing problem of introduced skilled workers. In terms of fostering the R&D capital flow, the government should strengthen exchanges and cooperation in innovative activities among regions. For example, efforts should be made to incentivize cross-regional applications for open topics in national key laboratories and joint applications by researchers from different regions for various levels and types of projects, including national and provincial natural science funds and social science funds. (2) Enhancing the ability of regions to absorb, transform, and utilize external knowledge, the moderating effect of regional absorptive capacity can be unleashed. This can be accomplished from three perspectives. Firstly, since education and human capital are important influencing factors of absorptive capacity, we can improve regional absorptive capacity by increasing investment in education, improving the quality of education, and enhancing human capital development, such as establishing high-level universities, strengthening the construction of teaching staff, and improving the education and teaching system. Secondly, the government should offer more favorable tax policies for scientific research and provide more R&D funding support for enterprises, and encourage their R&D investment. According to the dual role of enterprise R&D, this helps to improve the absorptive capacity of the enterprise, which in turn helps to strengthen the basic ability of the region to absorb foreign knowledge. Thirdly, efforts should be made to further improve transportation and information infrastructure, enhance interregional population, knowledge, and trade connections, creating more opportunities for knowledge inflow, thereby enhancing the region’s ability to identify and absorb knowledge.

”

Please see it on pages 42 to 44 of the manuscript. 

Thank you again for your valuable guidance and feedback.

Kind regards.

Comment 3. Manuscript need editing, please ensure to edit the manuscript.

Response 3: We have made corresponding edits to the article to ensure its logic and fluency.

Warm regards.

Comment 4. The conclusion should be extended, and contribution should be justified.

Response 4: We have rechecked and improved the conclusion section of this article, and the specific modifications are as follows:

We have added a sentence on page 41, which reads:

“

In 2008, 2015, and 2021, the absorptive capacity of the Eastern region measured 1.75, 3.55, and 5.75, respectively. Over the same years, the absorptive capacity of the Central region were 0.68, 1.53, and 2.43, respectively and those of the western region were 0.51, 1.55, and 2.59, respectively. The significant difference in regional absorptive capacity in China suggests that the effect of unit R&D factor inflows on regional economic growth will vary depending on regional absorptive capacity.

”

We have added a sentence on page 42, which reads:

“

The direct and indirect effects of R&D personnel inflow accounted for 12.121% and 87.878% of the total effect, respectively. Meanwhile, the direct and indirect effects of R&D capital inflow accounted for 9.302% and 89.583% of the total effect, respectively.

”

This sentence extends the previous sentence.

We have added a sentence on page 42, which reads:

“

Fourth, from the spatial perspective, the moderating effect of regional absorptive capacity is significant.

”

We believe that this helps to serve as the starting sentence of our conclusion#4.

Moreover, we have added a sentence in the penultimate sentence of Section 6, which reads:

“

According to the conclusions above, we find that enhancing regional absorptive capacity is equally important as facilitating the flow of R&D factors.

”

Kind regards.

Comment 5. Contribution need to be extended.

Response 5: This comment is valuable. We have extended the contribution section of this article, and the details are as follows:

We change the starting sentence of this paragraph, which reads: 

“

Our study is distinguished from other studies and makes contributions to the literature in the following aspects.

”

Please see it on page 9, the last paragraph of this section.

In addition, we have extended the contribution on page 9, which reads:

“

In the study of Bai, et al. [9], authors have regarded that the knowledge brought by the flow of R&D factors can automatically lead to spillovers. They have not considered the impact of a region's knowledge absorptive capacity on knowledge spillover effects. This means that the spillover effects brought by the inflow of unit factors are the same for all regions, but in fact, they are different.

”

We sincerely thank you again for your valuable comments, which have been very helpful in improving our article.

Besh wishes!

Wishing you happiness every day!

---

## [Decision Letter · Decision Letter 2]

5 Aug 2024

PONE-D-23-25669R2The impact of R&D factors flow and regional absorptive capacity on China’s economic growth: Theory and evidencePLOS ONE

Dear Dr. Liang,

Thank you for submitting your manuscript to PLOS ONE. After careful consideration, we feel that it has merit but does not fully meet PLOS ONE’s publication criteria as it currently stands. Therefore, we invite you to submit a revised version of the manuscript that addresses the points raised during the review process. **Comments from the editorial office**: During our evaluation of the manuscript, we came across the following article by Wan et al., titled, 'The impact of R & D elements flow and government intervention on China’s hi-tech industry innovation ability' (https://doi.org/10.1080/09537325.2021.1988554), which looks at a very similar question. Please note that PLOS ONE has been specifically designed for the publication of the results of original research contributing to the base of academic knowledge (https://journals.plos.org/plosone/s/criteria-for-publication#loc-1). In this case, could you please clearly state how your submission advances on the research reported by Wan et al., in the publication stated above? Please also note that submissions that are  similar to previous work must include a sound scientific rationale for the submitted work and clearly reference and discuss the existing literature (http://journals.plos.org/plosone/s/criteria-for-publication#loc-2).   

We look forward to receiving your revised manuscript.

Kind regards,

Annesha Sil, Ph.D.

Associate Editor

PLOS ONE

Journal Requirements:

Reviewers' comments:

Reviewer's Responses to Questions

**Comments to the Author**

1. If the authors have adequately addressed your comments raised in a previous round of review and you feel that this manuscript is now acceptable for publication, you may indicate that here to bypass the “Comments to the Author” section, enter your conflict of interest statement in the “Confidential to Editor” section, and submit your "Accept" recommendation.

Reviewer #1: All comments have been addressed

Reviewer #3: All comments have been addressed

2. Is the manuscript technically sound, and do the data support the conclusions?

Reviewer #1: Yes

Reviewer #3: Yes

3. Has the statistical analysis been performed appropriately and rigorously? 

Reviewer #1: Yes

Reviewer #3: Yes

4. Have the authors made all data underlying the findings in their manuscript fully available?

Reviewer #1: Yes

Reviewer #3: No

5. Is the manuscript presented in an intelligible fashion and written in standard English?

Reviewer #1: Yes

Reviewer #3: Yes

6. Review Comments to the Author

Reviewer #1: (No Response)

Reviewer #3: As authors have followed my suggestions, I am pleased to recommend this study for acceptance.

1. Based on my suggestions, authors have improved the abstract by providing a background, including details about the data and sample, and providing brief policies.

2. The authors have also improved the introduction of the study by expanding the contribution of the study.

3. They have also improved the data and methodology section.

4. I have also noticed some improvements in the policy directions of the study.

Summing up, the article has improved and I thereby, recommend it for publication.

7. PLOS authors have the option to publish the peer review history of their article (what does this mean?). If published, this will include your full peer review and any attached files.

Reviewer #1: No

Reviewer #3: No

---

## [Author Response · Author response to Decision Letter 2]

7 Aug 2024

Response to the editorial office

Letter to the editorial office

Dear the editorial office,

We sincerely thank you for your thorough review of our manuscript. We have noted your concerns about similarity and fully understand them. Based on your suggestion, we have included an overview of Wan et al.'s article in the introduction section of our paper and conducted some discussion. Our article differs significantly from Wan et al.'s article. Below, we will elaborate on the differences between our work and the Wan et al. article you mentioned.

Response to the editorial office:

Although both Wan et al and our team study R&D factor flow (also referred to as R&D elements flow), and both of us use R&D factor flow as the independent variable in our empirical tests, our articles differ significantly and focus on different aspects. The details are as follows:

(1) In terms of research content, Wan et al. study the impact of R&D factor flow on the innovation capabilities of China's hi-tech industries, as well as the moderating role of government intervention between R&D factor flow and the innovation capabilities of China's hi-tech industries. In contrast, our research focuses on the impact of R&D factor flow on regional economic growth in China, and the moderating role of regional knowledge absorptive capacity between R&D factor flow and China's economic growth.

(2) In terms of research subjects, Wan et al. use the innovation capabilities of China's hi-tech industries as the dependent variable, whereas we use China's economic growth as the dependent variable. Wan's study focuses on the industry level, while our research is conducted at the regional level. We believe our study is broader and more macro in scope.

(3) In terms of research focus, Wan et al emphasize government intervention and use it as the moderating variable in their study, whereas we focus on regional knowledge absorptive capacity and use it as the moderating variable in our paper. One of the innovations of our research is that existing literature on R&D factor flow and economic growth does not consider the impact of regional knowledge absorptive capacity. Our study addresses this gap. We propose and demonstrate that the contribution of unit innovation factor inflow to regional economic growth varies depending on the regional knowledge absorptive capacity.

(4) In terms of research methods, in the theoretical analysis section, Wan et al use narrative descriptions to explain the impact of R&D factor flow and government intervention on the innovation capabilities of China's hi-tech industries. However, we not only use narrative descriptions but also employ mathematical models to provide mathematical proof, illustrating the impact of R&D factor flow and regional knowledge absorptive capacity on China's regional economic growth.

(5) In terms of research methods, in the empirical study, Wan et al. simply used the proportion of government funds to R&D funds as a proxy variable for government intervention. However, we constructed a comprehensive index system to measure regional knowledge absorptive capacity. We presented the measurement results of the knowledge absorptive capacity of different provinces in China, which helps readers analyze and study the knowledge absorptive capacity across various Chinese provinces.

(6) In terms of research methods, in the empirical study, Wan et al. employed the SAR model from spatial econometrics for the analysis. In contrast, we utilized the SDM model from spatial econometrics for our analysis. Renowned spatial economist Elhorst has noted that the advantage of the SDM model is its ability to simultaneously capture both spatial lag and spatial error effects, providing a more accurate description of the relationships in spatial data. The SDM model can capture the spatial spillover effects of the independent variable (R&D factor flow) on the dependent variable, which the SAR model cannot. The importance of studying the spatial spillover effects of R&D factor flow lies in the fact that R&D factor flow benefits not only the economic growth of the recipient region but also the economic growth of neighboring regions, which the SAR model cannot demonstrate.

(7) In addition, Wan et al.'s study did not conduct a heterogeneity analysis, whereas we also performed a heterogeneity analysis. Therefore, our content is more comprehensive.

(8) Finally, it is important to point out the issue of keyword translation. We chose to translate it as "R&D factor flow" rather than "R&D elements flow" because we believe that "factor" is more aligned with the common economic terminology for production factors in English. Of course, as native speakers, you may have a more accurate judgment on this matter, and I am open to your suggestions.

Based on your concerns and valuable suggestions, we have added a brief overview of Wan et al.'s article to the introduction section of our manuscript:

“

Wan, et al. [12] argued that both R&D personnel flow and R&D capital flow have significantly promoted China's hi-tech industry innovation capabilities, while government intervention exerts a negative regulatory impact between R&D factor flow and hi-tech industry innovation capabilities [12].

”

See lines 120 to 123 in the introduction section of our manuscript.

To facilitate a concise comparison, we have added the following sentences to the contributions section of the introduction in our manuscript:

“

Fourth, different from Wan et al.’s [12] study, which focuses on the impact of R&D factors flow and government intervention on innovation capabilities and employs the spatial autoregressive (SAR) model to demonstrate this [12], our research uses the SDM model to examine the relationships between R&D factors flow, regional absorptive capacity, and China's economic growth. Methodologically, the SDM model allows for the decomposition of the direct effects and spatial spillover effects of the independent variable (R&D factors flow), leading to more accurate spatial estimates compared to the SAR model.

”

See lines 195 to 201 in the introduction section of our manuscript.

Regarding the references, we have updated Wan et al.'s article from [35] in the old manuscript to [12] in the new manuscript, resulting in corresponding adjustments to the other reference numbers.

We assure you that our manuscript is original and innovative. If you have any further questions, please feel free to reach out to us, and we will be happy to address them.

Warm regards,

Corresponding Author

Appendix:

See the table in the document

---

## [Editor Report · Decision Letter 3]

2 Sep 2024

The impact of R&D factors flow and regional absorptive capacity on China’s economic growth: Theory and evidence

PONE-D-23-25669R3

Dear Dr. Liang,

We’re pleased to inform you that your manuscript has been judged scientifically suitable for publication and will be formally accepted for publication once it meets all outstanding technical requirements.

Kind regards,

Laura Kelly

Division Editor

PLOS ONE
---

## [Editor Report · Acceptance letter]

12 Sep 2024

PONE-D-23-25669R3 

PLOS ONE

Dear Dr. Liang, 

I'm pleased to inform you that your manuscript has been deemed suitable for publication in PLOS ONE. Congratulations! Your manuscript is now being handed over to our production team.

Kind regards, 

on behalf of

Dr. Laura Hannah Kelly 

Staff Editor

PLOS ONE